# Holistic bursting cells store long-term memory in auditory cortex

Ruijie Li [1,2], Junjie Huang[3,4], Longhui Li[3], Zhikai Zhao [3], Susu Liang [3], Shanshan Liang [2], Meng Wang[3], Xiang Liao [3], Jing Lyu [5], Zhenqiao Zhou [5], Sibo Wang [5], Wenjun Jin[2,6], Haiyang Chen[7], Damaris Holder [4], Hongbang Liu[1], Jianxiong Zhang[2], Min Li [5], Yuguo Tang[5], Stefan Remy [4,8], Janelle M. P. Pakan [8,9,10] ✉, Xiaowei Chen [2,6] ✉ & Hongbo Jia [1,4,5,11] ✉

The sensory neocortex has been suggested to be a substrate for long-term memory storage, yet which exact single cells could be specific candidates underlying such long-term memory storage remained neither known nor visible for over a century. Here, using a combination of day-by-day two-photon $Ca^{2+}$ imaging and targeted single-cell loose-patch recording in an auditory associative learning paradigm with composite sounds in male mice, we reveal sparsely distributed neurons in layer 2/3 of auditory cortex emerged step-wise from quiescence into bursting mode, which then invariably expressed holistic information of the learned composite sounds, referred to as holistic bursting (HB) cells. Notably, it was not shuffled populations but the same sparse HB cells that embodied the behavioral relevance of the learned composite sounds, pinpointing HB cells as physiologically-defined single-cell candidates of an engram underlying long-term memory storage in auditory cortex.

The search for the cellular embodiment of long-term memories in the brain has been carried out for more than a century since the original theoretical proposition of the 'engram'[1,2]. Recent reviews[3–7] of historical and modern literature have summarized various cellular embodiments of memory engrams that meet at least one of the four defining criteria: 'content', 'persistence', 'ecphory', and 'dormancy'. To date, extensive investigations into memory engrams have been carried out in the hippocampus[8–10], the amygdala[11,12], the prefrontal, and other higher cortical areas[13,14] as well as the motor cortex[15] and collective sets of those brain regions[16] and beyond[17]. In this line of research, molecular engram tagging methods based on immediate early gene (IEG) expression have been developed and applied extensively[4,7,18]. However, efforts to identify memory engrams in the sensory neocortex using IEG tagging methods have yielded conflicting results. For example, fos-tagged neurons in the superficial layers of mouse barrel cortex through a whisker-dependent associative learning task did not represent the specific learning experience[19].

Complementary to the IEG approach, cumulative physiological evidence has demonstrated specific and persistent long-term plasticity in the sensory neocortex, particularly including auditory cortex

[1]Advanced Institute for Brain and Intelligence and School of Physical Science and Technology, Guangxi University, Nanning 530004, China. [2]Brain Research Center and State Key Laboratory of Trauma, Burns, and Combined Injury, Third Military Medical University, Chongqing 400038, China. [3]Center for Neurointelligence, School of Medicine, Chongqing University, Chongqing 400030, China. [4]Leibniz Institute for Neurobiology (LIN), 39118 Magdeburg, Germany. [5]Brain Research Instrument Innovation Center, Suzhou Institute of Biomedical Engineering and Technology, Chinese Academy of Sciences, Suzhou 215163, China. [6]Chongqing Institute for Brain and Intelligence, Guangyang Bay Laboratory, Chongqing 400064, China. [7]State Key Laboratory of Molecular Developmental Biology, Institute of Genetics and Developmental Biology, Chinese Academy of Sciences, 100101 Beijing, China. [8]Center for Behavioral and Brain Science (CBBS), Otto von Guericke University, 39120 Magdeburg, Germany. [9]Institute for Cognitive Neurology and Dementia Research, Otto von Guericke University, 39120 Magdeburg, Germany. [10]German Center for Neurodegenerative Diseases (DZNE), 39120 Magdeburg, Germany. [11]Institute of Neuroscience and the SyNergy Cluster, Technical University of Munich, 80802 Munich, Germany. ✉e-mail: janelle.pakan@med.ovgu.de; xiaowei_chen@tmmu.edu.cn; hongbo.jia@lin-magdeburg.de

(AuC)[20–22]. For example, fear conditioning produces sound frequency-specific, cholinergic-dependent receptive field changes in AuC[20,23,24]. Sensory cortical neurons are also known to encode behaviorally relevant sensory stimulus information[25,26]. Moreover, sensory experience-dependent behavioral choices can be entrained or manipulated by artificially exciting specific ensembles of sensory cortical neurons[26–30]. However, most of these studies lack a single-cell traceable history of changes in physiological activity over the memory formation process. As such, the long sought-after single-cell embodiment of a long-term memory trace in the sensory neocortex remains elusive. To tackle this challenge, we applied a combination of chronic two-photon neuronal population Ca²⁺ imaging[31] and single-cell targeted loose-patch recording[32,33] in the supragranular layer (layers 2/3, L2/3) across a stimulus-reward associative learning paradigm. This combined method offers both high temporal precision and a single-cell-traceable history, allowing us to identify physiological embodiments of a long-term memory trace in the sensory neocortex with single-cell precision.

In the previous study[34], we have established an auditory associative learning paradigm using composite multi-tone chords as reward-associated stimuli. The specificity of chord composition led to identifying so-called 'holistic bursting' cells in L2/3 of auditory cortex in trained animals, which strongly responded with bursting to a specific learned chord but not to single component tones. This interesting observation guided us to a deeper dive here in this study, as we devised a series of experiments to test whether the same class of physiologically defined 'holistic bursting' cell meets each of the engram-defining criteria. To do so, we first reproduced our previous finding but with an improved experimental scheme, using day-by-day two-photon Ca²⁺ imaging to track the responsiveness of each imaged single cell in each day instead of a 2-timepoint report at only the naive and the well-trained stages. This revealed a new insight, that sparse single neurons show a step-wise transformation from a status of quiescence into a status of bursting responsive, demonstrating that the existence of 'holistic bursting' cells observed on the last day of learning (as previously reported[34]) is likely a result of the learning progress. Thus, this fulfills the first defining criterion of an engram, 'content', i.e., the engram is not only a representation of the specific content of memory but also a result of the memory formation process. From this point on, we further tested each of the other three engram-defining criteria ('persistence', 'dormancy', 'ecphory') on these specific bursting cells in auditory cortex.

## Results

### Voluntary auditory associative learning with two composite sounds

Head-fixed mice were trained with the presentation of a brief sound stimulus (duration 50 ms), after which a droplet of water was dispensed (100 ms following the end of the sound) to form a sound-associated licking behavior[34,35]. Two specific 4-tone composite chords for training, which were both novel to the animals, were associated with two different water dispense spouts, chord 1 resulting in water dispensed from the right and chord 2 from the left (Fig. 1a, Methods). The animals were freely allowed to lick at any time, and no punishment was applied in any condition, but water was dispensed strictly upon the sound stimulus and regardless of the animal's behavior. For the analysis of behavioral performance, a trial was defined as a success if the animal's first initiated lick touched the correctly assigned spout within 1000 ms following the sound stimulus onset. Thus, unlike commonly studied 2-choice tasks[36] where the sensory stimuli to be learned have already been extensively presented in a pre-training course before the sensory-behavior association, our protocol simultaneously examined sound-associated behavior and sound-evoked neuronal responses from a naive state (day 1, nearly 0% behavioral performance, Fig. 1a, $4.62 \pm 2.05\%$ for chord 1 and $3.85 \pm 2.77\%$ for chord 2, $N = 13$ mice, mean $\pm$ sem) and in each training session (1 session per day) until the

animals reached a high performance of nearly 100% (day 6, $100 \pm 0\%$ for chord 1, $P = 6e{-}15$; $100 \pm 0\%$ for chord 2, $P = 2e{-}13$, Two-sided paired t-test).

### Day-by-day monitoring and verification of 'holistic bursting' cells

Single neuronal responses in AuC L2/3 evoked by each of the training chords were tracked in each training session (1 session per day) by using chronic two-photon Ca²⁺ imaging with GCaMP6m[31] (e.g., Fig. 1b). To calibrate the Ca²⁺ response, on the last day of training (day 6), the glass of the cranial window was removed, and we acutely performed targeted single-cell loose-patch recordings simultaneously with two-photon imaging in the awake state (e.g., Fig. 1c). We obtained a calibration curve with an extended dataset including cells of a broad range of responsiveness ($N = 17$ cells pooled from 12 animals, Supplementary Fig. 1), by which, a threshold of $\Delta f/f \geq 0.8$ was defined for identifying Ca²⁺ response events as a burst firing event (within the analysis time window of 500 ms from sound stimulus onset). In a subset of loose-patch recording experiments, the responses to individual constituent pure tones (i.e., those tones that together make up each defined chord) were also tested (Fig. 1d, Methods, note that individual tones were never presented through learning but only tested on the last day). Notably, the number of spikes in a burst firing event evoked by the preferred chord was significantly greater than the arithmetic sum of the number of spikes evoked by the four corresponding constituent tones individually presented at an equalized loudness (Fig. 1d). These experiments (Fig. 1c, d, Supplementary Fig. 1a, c, e) reproduced our previously reported acute observation of 'holistic bursting' (HB) cells in animals that were already well-trained[34]. However, in the improved chronic experimental scheme here in this study, our directly targeted loose-patch recording following day-by-day two-photon Ca²⁺ imaging precisely pinpoints that these specific single cells indeed have acquired such unique bursting response property through the learning progress.

In view of the decades-long accumulated physiological knowledge of neuronal burst firing as an intrinsic firing mode that is highly relevant for information transmission fidelity and synaptic plasticity[37,38], what remains unclear here is whether the bursting firing evoked by a learned chord is different than spontaneous bursting activities. Here, under the guidance of two-photon Ca²⁺ imaging (GCaMP6m), we specifically targeted either HB cells or non-HB cells and recorded both sound-evoked activity and spontaneous activity (Fig. 1e, f, see Methods for detail). In both groups of non-HB cells and HB cells, the spontaneous bursting event occurrence rate (in the absence of a sound stimulus, Fig. 1g) was comparably low (non-HB cells: 0.04\0.02–0.06 Hz, $N = 9$ cells pooled from 5 animals, HB cells: 0.025\0–0.0775 Hz, $N = 14$ cells pooled from 10 animals, $P = 0.775$, Two-sided Mann–Whitney test). In contrast, the probability of a bursting response evoked by the specific learned chord in HB cells was almost 100% (100%\87.5–100%, $N = 14$ cells, Fig. 1h). Moreover, the number of spikes of a learned chord-evoked burst event (8\5–12.75, $N = 235$ events pooled from 14 cells) was significantly more than that of a spontaneous burst event in either non-HB cell (3\3–4, $N = 79$ events pooled from 9 cells, ****$P = 7.5e{-}11$) or HB cell (4\3–7, $N = 60$ events pooled from 14 cells, ****$P = 3.1e{-}5$, one-way ANOVA test and Tukey post-test, Fig. 1i). Last but not least, the instantaneous firing rate of a learned chord-evoked burst event (72.44\49.24–109.9 Hz, $N = 235$ events pooled from 14 cells) was also significantly higher than that of a spontaneous burst event in either non-HB cells (46.01\21.91–80.65 Hz, $N = 79$ events pooled from 9 cells, ****$P = 3.8e{-}5$) or HB cells (58.82\33.52–83.78 Hz, $N = 60$ events pooled from 14 cells, *$P = 0.0344$, one-way ANOVA test and Tukey post-test, Fig. 1j). Taken together, the intense bursting responses of HB cells convey high-fidelity information of the learned composite sounds and distinguish themselves from the spontaneous bursting of either non-HB cells or themselves.

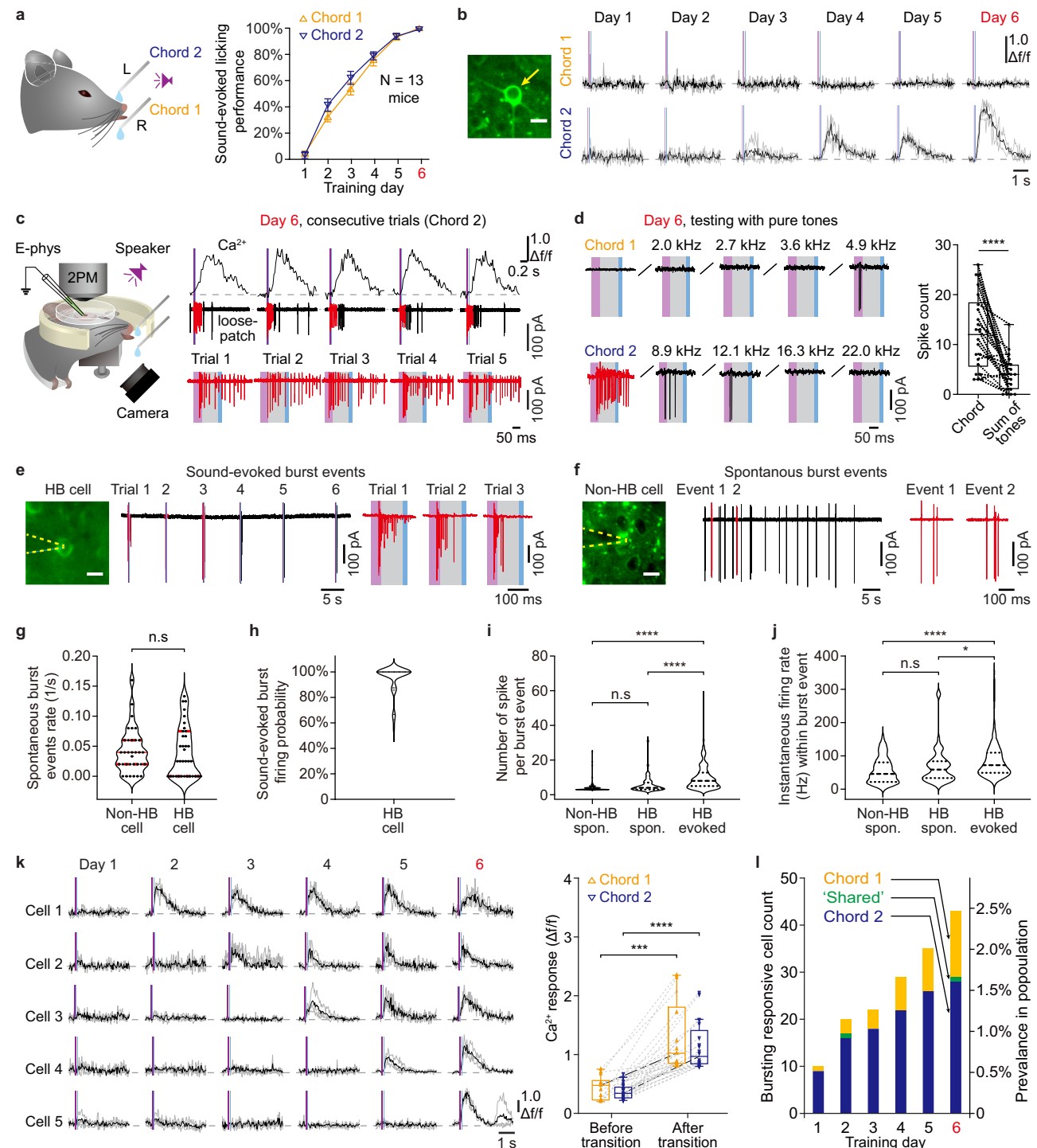

## HB cells emerged from quiescence in a stepwise manner through learning

The calibrated chronic Ca²⁺ imaging enabled functional tagging of individual cells, i.e., each single cell's sound-evoked response level as reported by time-resolved Ca²⁺ fluorescence imaging, analogous to the expression level of IEGs as reported by static fluorescence imaging. Those HB cells tagged on the last day of the training course (day 6) were particularly interesting as the bursting responsiveness was unique to a specific learned chord. We then asked if these cells were already naively burst-responsive to the chords to be learned from the beginning of learning, or if they gained de novo bursting responsiveness to the chords through the course of learning. A retrospective analysis of the population of all chronically imaged cells (1862 neurons,

pooled from 13 animals) showed that, for a cell that was quiescent on day 1 and identified as bursting responsive on day 6, a stepwise transition from quiescence to bursting occurred across days 2 to 6 (Fig. 1k). Notably, there existed sparse cells which were identified as bursting to a chord on day 1, but many of these had lost their chord-evoked bursting responsiveness during the subsequent training days (5/6 cells for chord 1, 8/17 cells for chord 2, for examples see Supplementary Fig. 2). As such, the quiescence-emerging bursting neurons constituted a major fraction of the HB cells tagged on day 6 (Fig. 1l, chord 1: 13/14 = 92.9% cells, 95% CI 49.4–100%; chord 2: 20/29 = 69.6% cells, 95% CI 42.1–100%). Thus, our single-cell and day-by-day traceable two-photon imaging dataset provides evidence that the HB cells tagged on the last day of learning had mostly (33/43) gained de novo bursting

**Fig. 1 | Holistic bursting (HB) cells emerged from quiescence in a stepwise manner through learning. a** Sound-licking associative learning task (left) and learning curve (right). Chord 1 (2.0, 2.7, 3.6, 4.9 kHz) associated with water from right side; chord 2 (8.9, 12.1, 16.3, 22.0 kHz) associated with water from left side. Data with error bars are presented as the mean ± sem. Credits of Ms. Jia Lou. **b** Left: Two-photon image of an example neuron in AuC L2/3 labeled with GCaMP6m (cropped from whole image, scale bar: 10 μm, same arrangements for all example cells hereinafter). Right: Sound-evoked $Ca^{2+}$ signals of the example neuron day by day (trial-averaged traces in black overlayed with single trials in gray). 33 neurons of 1862 imaged neurons in 13 mice. were repeated independently with similar results. **c** Simultaneous two-photon $Ca^{2+}$ imaging and loose-patch recording of the same example neuron in panel b on day 6. Upper: $Ca^{2+}$ signals and corresponding loose-patch responses in 5 consecutive trials. Lower: magnified view of the loose-patch recordings. Red sections of traces identified as bursting events (same hereinafter). Purple shade: sound stimulation, blue shade: water dispense, same for all figures hereinafter. 2PM: two-photon microscope. Credits of Ms. Jia Lou. **d** Left: Targeted loose-patch recordings of a neuron that emerged with high $Ca^{2+}$ responsiveness to chord 2 through learning. Right: Box-whisker plots (boxes: median & 25–75% percentiles, whiskers: 10–90% percentiles, same convention applies to all boxplots hereinafter) 12\5.5–18 spikes per chord response, 4\1–6 per sum of 4 constituent tones response. $N = 22$ trials (pooled, 4 HB cells recorded in 3 mice), ****$P = 2.2e{-}05$, $t_{21} = 5.437$, Two-sided paired t-test. **e** A session of loose-patch recording of an example HB cell including consecutive sound-evoked response events. 235 events pooled from 14 cells were repeated independently with similar results. **f** A session of loose-patch recording of an example non-HB cell, spontaneous activity in the absence of sound stimuli. 79 events pooled from 9 cells were repeated

independently with similar results. **g** No significance of spontaneous burst events rate (1/s) between non-HB cells and HB cells. Burst firing rate: non-HB cells, 0.04\0.02–0.06 Hz; HB cells, 0.025\0–0.0775 Hz. n.s., $P = 0.7550$, Mann–Whitney U: 552, 9 non-HB cells in 5 mice, 14 HB cells in 10 mice. Two-sided Mann-Whitney test. **h** Distribution of sound-evoked burst firing probability of HB cells. 100% \87.5–100% (median\25–75% percentiles), 14 HB cells in 10 mice. **i** Violin plots of number of spikes per burst event among non-HB cell spontaneous burst events, HB cell spontaneous burst events, and HB cell sound-evoked burst events. Non-HB spon. vs. HB spon., n,s, $P = 0.3213$; Non-HB spon. vs. HB evoked, ****$P = 7.5e{-}11$; HB spon. vs. HB evoked, ****$3.1e{-}5$, $F_{(2, 371)} = 28.18$; One-way anova test and Tukey post-test. **j** Violin plots of instaneous firing rate (Hz) within burst event among non-HB cell spontaneous burst events, HB cell spontaneous burst events, and HB cell sound-evoked burst events. Non-HB spon. vs. HB spon., n,s, $P = 0.4241$; Non-HB spon. vs. HB evoked, ****$P = 3.8e{-}5$; HB spon. vs. HB evoked, *$P = 0.0344$; $F_{(2, 372)} = 10.99$; One-way anova test and Tukey post-test. **k** Left: sound-evoked (chord 2) $Ca^{2+}$ signals of example cells emerged from quiescence to bursting (high $Ca^{2+}$, $\Delta f/f > 0.8$) responsiveness through learning. Right: summary of $Ca^{2+}$ responses before and after the transition for the quiescence-emerged cells. Chord 1, $\Delta f/f$ value before transition: 0.48\0.23–0.60, after transition, 1.02\0.85–1.90, $N = 13$ cells, ***$P = 2.7e{-}04$, $t_{12} = 5.082$; Chord 2, $\Delta f/f$ value before transition: 0.34\0.26–0.46, after transition, 0.97\0.84–1.47, $N = 20$ cells. ****$P = 1.1e{-}7$, $t_{19} = 8.242$, Two-sided paired t-test. Box-whisker plots: boxes represent 25% and 75% percentiles (Q1 and Q3), central bars represent the median, whiskers represent $Q1{-}1.5 \times IQR$ (interquartile range) and $Q3 + 1.5 \times IQR$. **l** Stacked histograms showing the number of cells exhibited bursting responsiveness in each training day for all cells tagged as bursting on day 6. Data pooled from altogether 1862 imaged neurons in 13 mice.

responsiveness as a unique physiological biomarker of the learned composite sounds ('content[5]' of the memory).

## HB cells invariably embodied the learned composite sounds

If instructed to look 'left', one may look left, or right, or do nothing, but the initial auditory recognition of the word 'left' per se would remain invariant regardless of any subsequent behavior. Accordingly, the sparse bursting output of HB cells was shown to be specific to the learned composite sounds, but unaffected by either behavioral motivation or outcome in an acute testing session in our previous study[34]. What would happen if the chord 2 sound (left) and chord 1 sound (right) were assigned to the opposite behavioral choices? Here, taking advantage of the chronic single-cell traceable imaging, we introduced a behavioral choice condition reversal (Fig. 2a) after the initial training course and monitored the responsiveness of the tagged HB cells on each day of the reversal training course.

On the first reversal day (day 7), the behavioral performance dropped down to ~50% but returned to almost 100% again through the 4 days of reversal learning (Fig. 2a). Note that both chords were associated with the same amount of water dispensed, but from reversed sources, and throughout the reversal training the animals maintained a high licking response rate of >90% per stimulus. Thus, in this equal-valued reversal training, change of behavioral performance reflected changes of the behavioral choice but not the valence of sensory stimuli. At the single-cell level, the majority of HB cells (29/43 = 67.4%, 95% CI 45.1–96.9%) tagged on the last day of initial training (day 6) persisted with bursting responsiveness to the same chord through the reversal learning. A minority of HB cells (14/43 = 32.6%, 95% CI 4.99–64.8%, $P = 0.0222$, chi-square test) lost bursting responsiveness, but in view of the entire population, the bursting cell subpopulation remained consistent with newly emerged bursting cells (11/1862 new versus 14/1862 lost, $P = 0.549$, chi-square test) through the course of reversal learning (Fig. 2b). Strikingly, in contrast to the 42/43 HB cells that showed specificity to a particular chord at the end of the initial training course, there were no cells (0/43, 95% CI 0–8.58%, $P < 0.0001$) in which the specificity of the bursting representation switched from one chord to the other throughout the reversal learning (Fig. 2b). This null reversal of the sound representation selectivity of HB cells was in stark contrast with the full reversal of behavioral choice.

Note that, the newly emerged bursting responsive cells through the reversal phase could also be interpreted as conveying the newly assigned behavioral association condition, nevertheless, the persisted specific bursting responsive cells suggested that the sound composition information stored in those cells through the initial learning process was invariant.

## HB cells persistently embodied the learned composite chords in remote recall

Next, a group of mice ($N = 8$) were used to investigate the properties of remote recall. After the initial course of learning, these mice rested in their home cage with neither restriction of water nor exposure to the learned sounds for up to a month (number of resting days randomly assigned to each animal: from 7 days to 1 month, water restriction applied 24 h prior to testing). Then, on the remote testing day, 10 of the 12 HB cells tagged on the last learning day (of total 323 imaged cells, 8 mice) exhibited bursting responsiveness to the same learned sound (Fig. 2c). Neither the $Ca^{2+}$ responses of tagged HB cells nor the behavioral performance showed any significant difference between the last learning day and the remote testing day (Fig. 2d). This remote recall test provided evidence that HB cells manifested a long-term memory engram. Together with behavioral reversal experiments, these data reveal a striking feature of HB cells as embodying the long-term invariability of learned sounds ('persistence[5]' of memory).

## HB cells embodied the behavioral relevance of learned composite sounds

Both the equal-valued choice reversal and the remote recall tests above demonstrated how the HB cells that emerged through the initial learning course could invariably embody the learned sounds that were relevant to the licking behavior. What remained unclear was whether the relevance of the behavior (not necessarily the actual choice of behavior) was embodied in HB cells. Thus, our next test involved a new group of animals that experienced the extinction of association between the learned sound and water reward. For this test, the learning scenario was simplified, it involved only one composite sound (broadband-noise, BBN, with a constant waveform in each trial) and one water dispense spout to be consistent also with our previous study[34]. In the initial course of training (Fig. 3a, 'initial training', day 1 to day 5), the

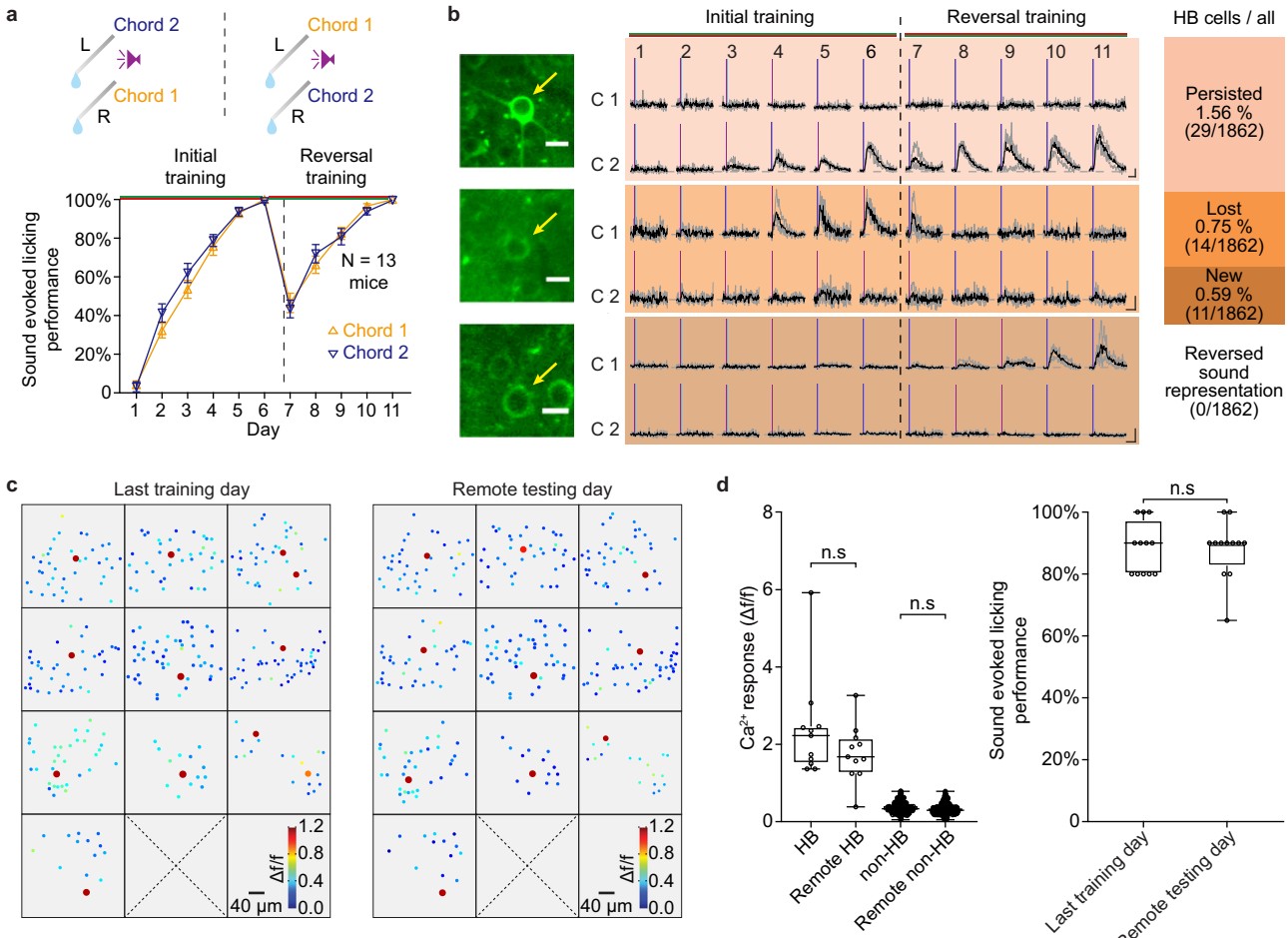

**Fig. 2 | Holistic bursting (HB) cells invariably and persistently embody the learned sounds. a** Upper: Schematic of initial training and reversal training. Lower: sound-evoked licking performance on each day. Day 7 (first day of reversal): 46.52 ± 4.97% for chord 1, 43.77 ± 4.6% for chord 2; day 11 (last day of reversal): 100% for both chords, $N = 13$ mice, mean ± sem, chord 1 (day7 vs. day11), $P = 3.9e−7$; chord 2 (day7 vs. day11), $P = 3.9e−8$, Two-sided paired t-test. L: left, R: right. Credits of Ms. Jia Lou. **b** Left to middle: two-photon images of 3 examples neurons (cropped from whole image, scale bar: 10 μm, the first cell is the same as in Fig. 1b) and sound-evoked $Ca^{2+}$ signals across all the 11 days. Scale bars: 1.0 Δf/f (vertical) & 1 s (horizontal). C1: chord 1. C2: chord 2, matching the subtypes summarized in the right column. 'Persisted': 29/1862 = 1.56% (95% CI 1.04–2.23%). 'Lost': 14/1862 = 0.75% (0.41–1.26%), $P = 0.0222$ versus 'Persisted'. 'New': 11/1862 = 0.59% (0.29–1.06%), $P = 0.55$ versus 'Lost'. 'Reversed': 0/42 = 0.0%, (0.0–8.8%), $P < 0.0001$ comparing to

28/42 = 66.67% the fraction of 'persisted' among HB cells by the end of initial training, chi-square test. **c** Pseudocolored heatmaps of $Ca^{2+}$ response (trial-averaged) of the same cells (323 cells from 10 FOVs of 8 mice) on the last training day and remote testing day (after 7 or more days testing in home cage), respectively. HB cells (Δf/f ≥ 0.8) are marked with a 2-fold larger dot for better visualization. **d** Comparison of $Ca^{2+}$ responses (left) and behavior performance (right) on the last training day and remote testing day. Left: HB cells: n.s., $P = 0.7891$, $t_{11} = 0.2741$, $N = 12$ cells; non-HB cells: n.s., $P = 0.5796$, $t_{310} = 0.5546$, $N = 311$ cells. Two-sided paired t-test. Right: Behavior performance: n.s., $P = 0.9046$, $t_{11} = 0.1227$, $N = 6$ mice. Two-sided paired t-test. Box-whisker plots: boxes represent 25% and 75% percentiles (Q1 and Q3), central bars represent the median, whiskers represent maximum and minimum.

timing of water delivery strictly followed the timing of the sound stimulus (100 ms after the sound end, Fig. 3a). Then, the sound and water were dissociated (Fig. 3a, 'dissociation', day 6 to day 9), i.e., water was not dispensed following the sound stimulus for each trial, but freely dispensed after the end of the entire session. As such, the animals no longer needed to perform sound-associated licking yet still obtained sufficient water reward in the same experimental context. Then, the same timing association between sound and water was established again (Fig. 3a, 're-association', day 10 to day 13). Behavior performance reached 100% through the initial training course, then drastically dropped to 0% through the course of dissociation, then recovered to nearly 100% through the re-association phase (Fig. 3b).

The definition of the HB cell still applies in this case, as the 'frozen' BBN could be physically regarded as a special case of composite sound as in our previous study[34]. At the population level, as expected, the prevalence of HB cells first rose through the initial training, then decreased through the dissociation, and finally increased again

through the re-association (Fig. 3c). However, the population analysis did not indicate whether or not the same single cells exhibited bursting responsiveness through each phase. To answer this question, a further single-cell analysis was carried out in an extended dataset (Fig. 3d). For the 30 HB cells tagged on the last day of initial training (day 5, out of all the 894 imaged cells pooled from 15 mice), a small proportion of them (4/30) endured with bursting response throughout the subsequent days including the dissociation phase (Fig. 3d, 'endured'). However, the majority of them (20/30) became dormant and did not exhibit sound-evoked bursting through dissociation but reinstated bursting responsiveness through re-association (Fig. 3d, 'reinstated'). The other small fraction (6/30) lost bursting responsiveness during dissociation and did not reinstate through re-association (Fig. 3d, 'lost'). In view of the entire imaged population, a few cells (2/894) newly emerged as HB cells through re-association (Fig. 3d, 'new'), comparable to the 'lost' ones (6/894, $P = 0.157$, chi-square test). These results strikingly demonstrate that it was the same sparse HB cells but not shuffled

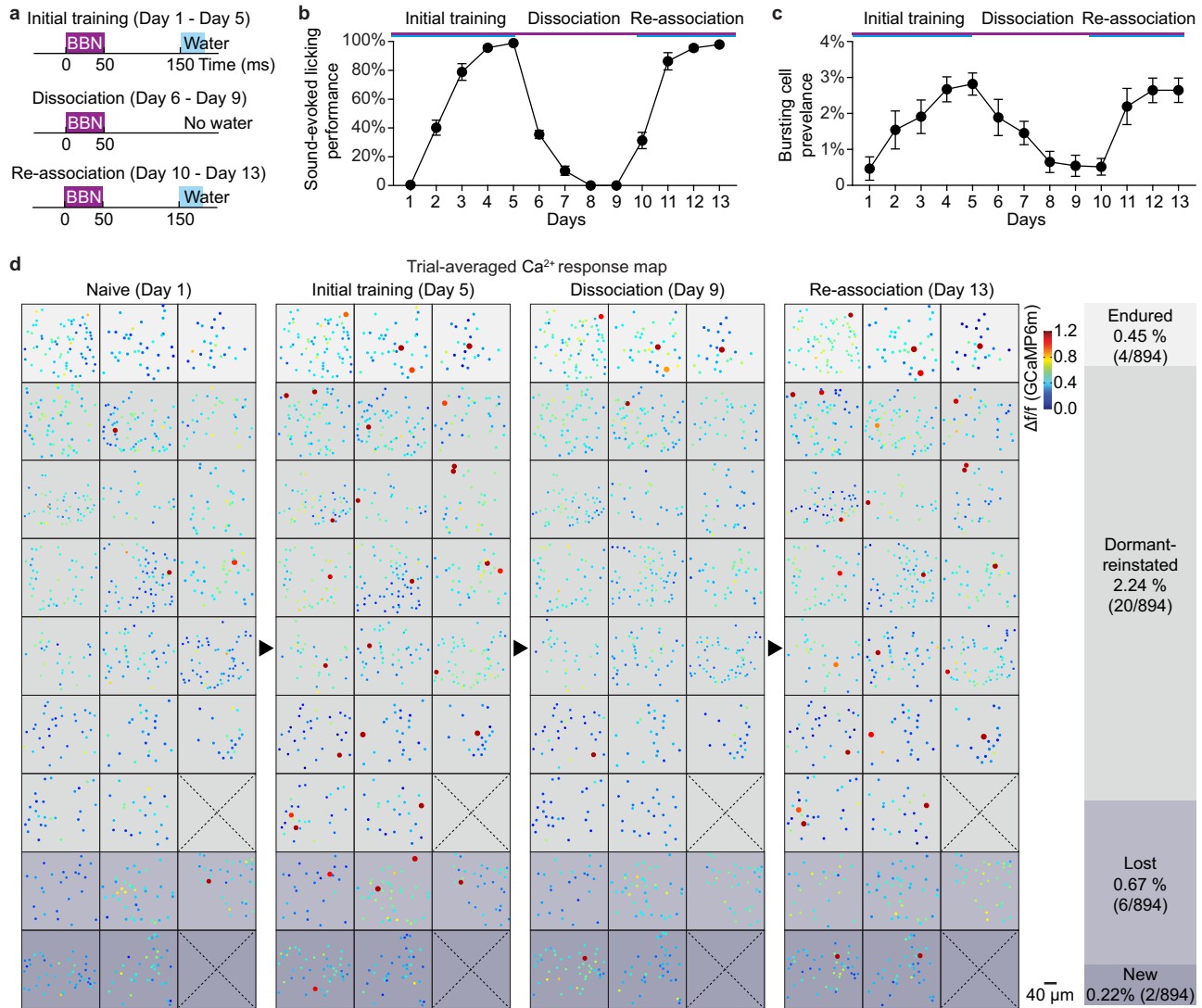

**Fig. 3 | Holistic bursting (HB) cells embody the behavioral relevance of learned sounds. a** Schematic of the experimental protocol for the dormancy test, including 3 phases: initial training (day 1 to day 5), dissociation (day 6 to day 9) and re-association (day 10 to day 13). BBN: Broad-band-noise (note: with a 'frozen' wave-form that does not change from trial to trial). **b** Sound-evoked licking performance (mean ± sem, $N = 10$ mice) in each day through the 3 courses (defined in panel **a**). **c** The prevalence of HB cells in each day (out of altogether 755 cells in 10 mice, mean ± sem). Day 1: 0.47 ± 0.32%; day 5 (end of initial training): 2.82 ± 0.31%, $P = 0.0004$; day 9 (end of dissociation): 0.55 ± 0.29%, day 13 (end of re-association):

2.65 ± 0.34%, $P = 0.0022$; Two-sided paired t-test. **d** Pseudocolored heatmaps of $Ca^{2+}$ response of the same cells (25 FOV in 15 animals, 894 cells, note that for this analysis only 4 timepoints are needed, thus more data from a different set of experiments are pooled together) from the beginning of learning (day1, 'naive') to the end of each course. HB cells (with response amplitudes $\Delta f/f \geq 0.8$) are marked with a 2-fold larger dot for better visualization. Shaded column on the right: summary of HB cell subtypes, 'endured': 0.45% (0.12–1.15%); 'dormant-reinstated': 2.24% (1.37–3.46%); 'lost': 0.67% (0.25–1.5%).

populations of AuC L2/3 cells that reinstated after dormancy to embody the behavioral relevance of the initially learned composite sound ('reinstated' vs 'endured': $P = 0.001$; vs 'lost': $P = 0.006$; vs 'new': $P = 0.0001$, chi-square test).

## HB cells are widespread throughout the auditory cortex irrespectively of tonotopic map

It has been well known that both classical and operant conditioning can lead to a global shift in the tonotopic map toward the frequency of the conditioned tone in the auditory cortex[20]. Instead of pure tones used in most previous studies, we trained the animals with either specifically composed multi-tone chords or BBN with a constant waveform. This is because each real-world behaviorally relevant sound has a unique characteristic composition of multiple tonal components, in contrast to the fact that single pure tones are rarely presented in naturalistic life. To relate this new learning paradigm with composite

sounds to classical studies with pure tones, we performed a new combined experiment with wide-field epifluorescence $Ca^{2+}$ imaging (large-view mapping, no single-cell resolution) and 2-photon $Ca^{2+}$ imaging (small view, single-cell resolution, and high sensitivity) in the same chord-trained animals sequentially (Fig. 4a, b, see Methods for details, both using the same GCaMP6m labeling). First, each of the learned chords possesses a cortical response zone that is largely overlapping with the response zones of their corresponding tone frequency band (chord 1: low tones 4 kHz & 8 kHz, overlap 66.5%\51.58–83.55%; chord 2: high tones 16 kHz & 32 kHz, overlap 78.69%\42.52–92.52%, $N = 8$ mice). Second, HB cells were found to be evenly and widely distributed throughout the auditory cortex in all the 3 major functional areas identified by classical means of tonotopy[39] (A1: 40%, A2: 28%, AAF: 11%, others: 21%, Fig. 4c, $N = 81$ HB cells), Third, nearly half (48%) of the identified HB cells are located beyond the response zones of either chord (Fig. 4d). Note that methodologically,

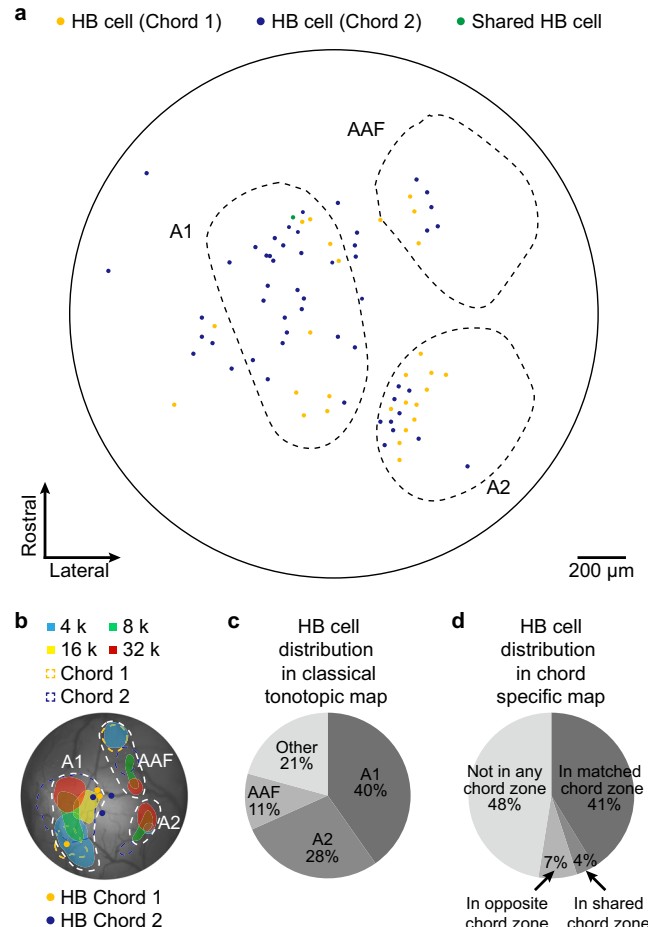

**a** HB cell (Chord 1) ● HB cell (Chord 2) ● Shared HB cell

**b** 4 k 8 k 16 k 32 k Chord 1 Chord 2
HB Chord 1 HB Chord 2

**c** HB cell distribution in classical tonotopic map
Other 21% A1 40% AAF 11% A2 28%

**d** HB cell distribution in chord specific map
Not in any chord zone 48% In matched chord zone 41% In opposite chord zone 7% 4% In shared chord zone

**Fig. 4 | Distribution of holistic bursting (HB) cells in auditory cortex.**
**a** Superposition of all identified HB cells followed by sequentially performed wide-field epifluorescence imaging (tonotopic mapping) and two-photon imaging (identifying HB cells). Dotted lines outline the boundaries of identified tonotopic map areas of A1 (Primary Auditory Cortex), A2 (Secondary Auditory Cortex) and AAF (Anterior Auditory Field). Note that the selection of two-photon imaging FOVs were evenly distributed in the circular cranial window (outer solid ring) across this cohort of 8 mice in order to minimize bias for the superposition of maps. **b** Example animal showing the relation between pure-tone response zones (sequentially tested tones: 4k, 8k, 16k, 32 kHz, labeled by colored shades as legend on graph) and the chord response zones acquired by wide-field imaging (yellow and blue dotted lines as legend on graph). The locations of HB cells identified in this animal are also overlayed. **c** Distribution of HB cells (total 81 cells) in the classical tonotopic map. **d** Distribution of HB cells in the chord specific map.

the epifluorescence low-resolution Ca²⁺ imaging collects a diffuse and integrated signal that mainly originates from the dense neurites (dendrites, axons) in the superficial cortical layers, thus reflects the input information flow to neuronal populations therein. In contrast, two-photon imaging pinpoints somatic Ca²⁺ signal, which reflects the output of specific single neurons. These data together support the notion that HB cells emerge with the specific bursting output property de novo through learning and are widely distributed in the auditory cortex regardless of the classical tonotopic organization.

### Establishing a pharmacological 'loss-of-function' intervention of HB cells in behaving animals
So far, multiple independent lines of evidence under different behavioral conditions (equal-valued choice reversal, remote recall, extinction & reassociation) suggest that HB cells in AuC L2/3 embody a long-term memory of learned, behaviorally relevant composite sounds, prompting us to probe a relation between the bursting activation of HB

cells and instantaneous memory readout. To this end, a test of sound-associated licking upon the inactivating manipulation of HB cells ('loss-of-function') is mandatory, because sound-associated licking events are almost always accompanied by bursting of HB cells bursts prior to licking initiation (Supplementary Fig. 3), but in contrast, HB cell bursts are not always accompanied by a subsequent licking action (see also Fig. 5 of our previous study[34]). Furthermore, spontaneously licking events occurring in the absence of sound stimuli are rarely accompanied by HB cell bursts (Supplementary Fig. 3). As HB cells are widely distributed throughout all primary and secondary auditory cortical areas (Fig. 4), we sought a pharmacological intervention that could act as a 'low-pass filter' of neuronal firing, i.e., suppressing high-frequency bursting neuronal output but not significantly suppressing non-bursting outputs in a sufficiently large yet limited volume of cortical tissue. Previous studies suggests that isoflurane exerts a very selective suppression effect on high-frequency activities at the order of magnitude of ~100 Hz[40,41] which is in the same range as the bursting of HB cells when evoked by learned sounds. Here, during gaseous isoflurane application as general anesthesia (Fig. 5a), HB cells were severely suppressed, and all lost their bursting responsiveness (Fig. 5b, c left, 9 of 9 HB cells). Notably, the majority of neurons (277/286) in the same imaging FOV as non-HB cells could still exhibit non-bursting responses to the same sound stimulus, and Bayes factor analysis[42] yielded 'absence of evidence' of the suppression effect on non-HB cells (Fig. 5b, c right). HB cells recovered bursting responsiveness 60 min after removal of the isoflurane in the inhalant (Fig. 5b, c left, 8 of 9 HB cells, $P = 0.81$, chi-square test), likewise, the sound-evoked licking behavioral performance also recovered to a level >90% at the same timepoint of imaging (Fig. 5d).

Therefore, this suggested that isoflurane could be mechanistically repurposed as a pharmacological intervention for targeted suppression of HB cells. To test this further without the generalized effects of the anesthesia, we devised a method of local aqueous application of isoflurane by means of rapid and precise nano injection of diluted isoflurane solution via a glass micropipette targeting AuC L2/3 (same pipette as used for single-cell electrophysiology, see Methods for details) in awake behaving mice (Fig. 5e). Neuronal responsiveness and behavior performance were tested within a short time window during which the intervention effect was relatively stable after a single bolus of injection (duration 10–20 s, injected volume estimated <1 nL, recording 1–20 min post injection). HB cells in the targeted AuC L2/3 area were suppressed and lost bursting responsiveness right after the injection, then later recovered bursting responsiveness at ~60 min post injection (Fig. 5f, g, all 9 of 9 HB cells). In contrast, Bayes factor analysis[42] yielded an 'evidence of absence' of suppression effect on non-HB cell responses in the same area (Fig. 5g). Moreover, this pharmacological intervention was controlled by normal brain fluid application and different concentrations of isoflurane in the injection solution, showing that the suppressing effect on the responsiveness of HB cells was isoflurane-dependent (Supplementary Fig. 4).

### Local isoflurane injection suppresses both bursting response and learned sound-associated behavior
The local application of aqueous isoflurane utilized the same biophysical mechanism of isoflurane that could preferentially suppress high-frequency bursting responses, yet with a reduced global network impact in contrast to inhalation of gaseous isoflurane. Interestingly, with unilateral AuC L2/3 microinjection of isoflurane under the two-photon microscope, the learned sound-associated licking performance was reduced only to ~50% and then fully recovered after 60 min post application (Fig. 5h). This partial reduction is plausible as HB cells for the same sound (chord 2 in this test) could exist in bilateral AuC L2/3. Thus, another group of animals was further tested for bilateral AuC L2/3 isoflurane injection, and their behavioral performance further dropped to a much lower level (Fig. 5h). Under the intervention of local

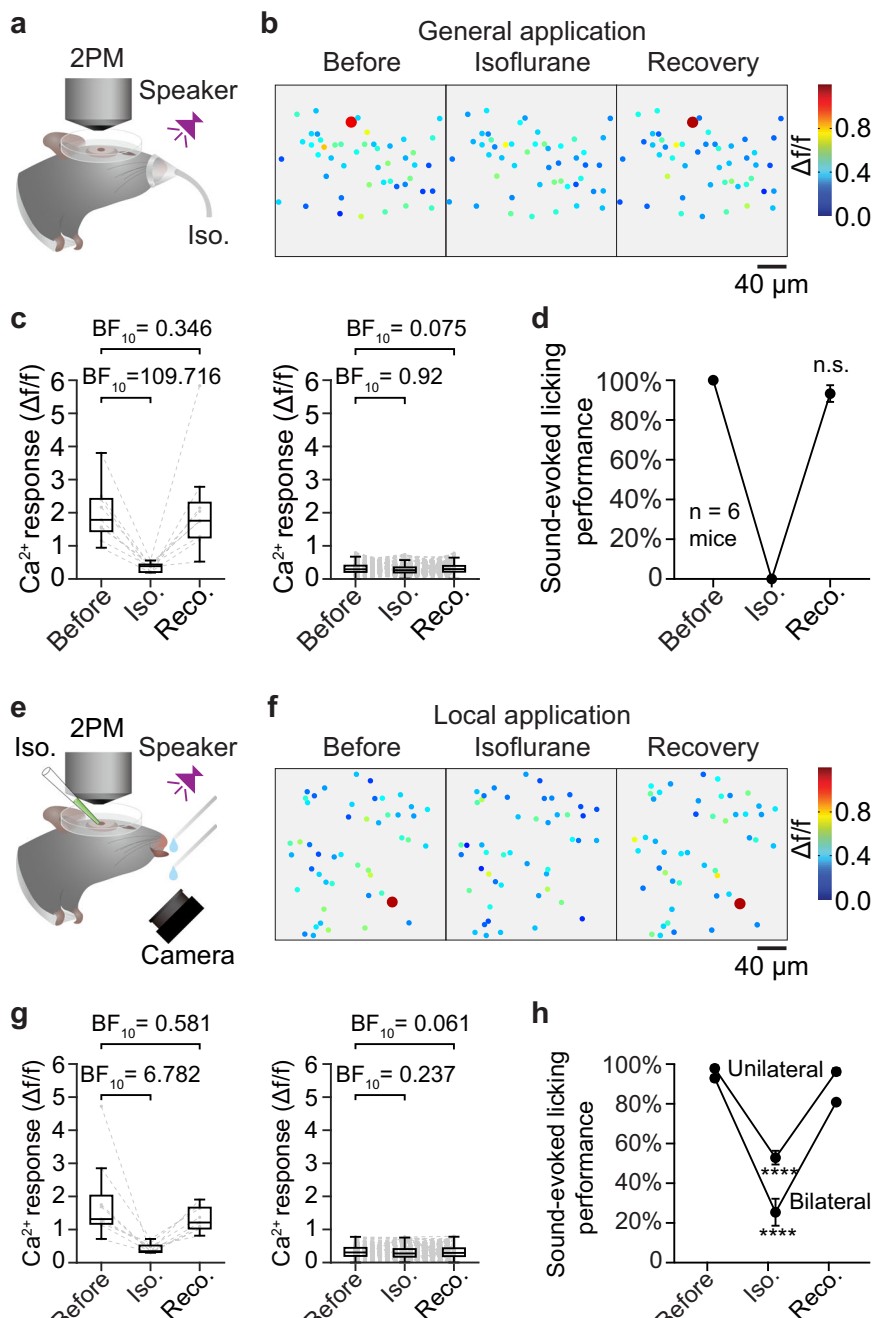

**Fig. 5 | Local isoflurane injection suppresses both bursting response and learned sound-associated behavior. a** Schematic of simultaneous two-photon Ca²⁺ imaging (2PM) and general isoflurane (Iso.) application. Credits of Ms. Jia Lou. **b** Color map of trial-averaged Ca²⁺ response (chord 2 stimulation, same for all panels herein) of the same cells before isoflurane application, post-application and after recovery. Bursting responsive cells (with response amplitudes $\Delta f/f \geq 0.8$) are marked with a 2-fold larger dot for better visualization. **c** Summary of Ca²⁺ responses of HB cells and non-HB cells under isoflurane (Iso.) intervention and after recovery (Reco.). Left: Summary of HB cells' Ca²⁺ responses, $\Delta f/f$ values, before: 1.784\1.537–2.389; iso: 0.385\0.209–0.424, $BF_{10} = 109.716$, error = 0.00006%; recovery: 1.758\1.277–2.146, $BF_{10} = 0.346$, error = 0.005%; $N = 9$ cells. Right: Non-HB cells, $\Delta f/f$ values: before: 0.293\0.212–0.402; iso: 0.266\0.196–0.35, $BF_{10} = 0.92$, error = 0.025%; recovery: 0.299\0.219–0.395, $BF_{10} = 0.075$, error = 0.0004%; $N = 277$ cells pooled from 6 mice ($BF_{10} < 1/3$: 'evidence of absence'; 1/3 < $BF_{10}$ < 3: 'absence of evidence'; $BF_{10}$ > 3: 'evidence of presence'). Bayesian paired samples t-test. boxes represent 25% and 75% percentiles (Q1 and Q3), central bars represent the median, whiskers represent Q1–1.5 × IQR (interquartile range) and Q3 + 1.5 × IQR. **d**, Comparison of sound-evoked licking performance before and after general isoflurane application. Before: 100 ± 0%, recovery: 93.33 ± 4.22%

(mean ± sem). Before vs. Recovery, n.s., $P = 0.1747$, $t_5 = 1.581$, $N = 6$ mice, Two-sided paired t-test. **e, f** Similar arrangement as panels **a** and **b** showing an example focal plane of local 2% isoflurane application (unilateral). Credits of Ms. Jia Lou. **g** Similar arrangement as panel **c** for 2% isoflurane local application. Left: HB cells, $\Delta f/f$ values, before: 1.315\1.185–1.749; iso: 0.346\0.321–0.49, $BF_{10} = 6.782$, error = 0.00009%; recovery: 1.212\1.033–1.636, $BF_{10} = 0.581$, error = 0.013%; $N = 9$ cells pooled from 8 animals. Right: Non-HB cells, $\Delta f/f$ values, before: 0.305\0.199–0.444; iso: 0.275\0.176–0.407, $BF_{10} = 0.237$, error = 0.093%; recovery: 0.291\0.194–0.432, $BF_{10} = 0.061$, error = 0.344%; $N = 358$ cells pooled from 8 animals. Bayesian Paired Samples T-Test. Box-whisker plots: boxes represent 25% and 75% percentiles (Q1 and Q3), central bars represent the median, whiskers represent Q1–1.5 × IQR (interquartile range) and Q3 + 1.5 × IQR. **h** Summary of sound-evoked licking performance under local isoflurane application. Unilateral application, before: 97.91 ± 1.43%, isoflurane: 52.88 ± 3.45%, recovery: 96.24 ± 1.72%, $N = 8$ mice, before vs. isoflurane: $P = 1e{-}10$, $t_{15} = 15.34$, before vs. recovery: $P = 0.1787$, $t_{15} = 1.411$, Two-sided Paired t-test; bilateral application, isoflurane: 25.42 ± 6.81%, $N = 6$ mice, before vs. isoflurane: $P = 8.5e{-}10$, $t_{23} = 9.941$, before vs. recovery: $P = 0.0522$, $t_{23} = 2.048$, Two-sided paired t-test.

aqueous isoflurane injection in AuC L2/3, the animals could still perform licking behaviours and the non-HB cells could still exhibit non-bursting responses to the same learned sound, but the temporary loss of HB cell bursting responses severely impaired the performance of the learned sound-associated licking. Our data demonstrate a vital role for HB cell specific sound-evoked bursting output in performing the learned sound-associated licking behavior, in an analogous manner to the 'loss-of-function' test involving IEG-tagged engram cells[7]. Note that the Ca$^{2+}$ imaging offered a readout of the net neuronal output in the imaged view in layer 2/3 and showed a specific suppression of the bursting response during local isoflurane application, mechanistically supported by literature[40,41] showing that isoflurane acts as a 'low-pass filter' of neuronal firing. Particularly, since the bursting of HB cells evoked by learned chords are among the strongest excitatory activations in L2/3 of AuC (Fig. 1e–j), they are also the most susceptible to local isoflurane injection. But we do not exclude the possibility that other specific neurons in deeper layers which may possess high-frequency bursting response capability (though not necessarily emerging through learning)[43] are affected by the spread of isoflurane and could also contribute to the reduction of learned associative behavior. Therefore, although the activity in deeper layers may affect the interpretation of this instantaneous isoflurane intervention experiment, the results of this experiment are also are consistent with the above-mentioned three lines of long-term experiments (Fig. 1, 'content'; Fig. 2, 'persistence'; Fig. 3, 'dormancy'), together, suggesting that HB cells are likely the specific candidates of cellular engrams underlying long-term memory storage in auditory cortex. To this point, we propose that the future development of an HB-cell-specific silencing technology will prove vital to further verify the current results with isoflurane suppression.

## Discussion

Here, we demonstrate that the combination of chronic (day-by-day) two-photon neuronal population Ca$^{2+}$ imaging and targeted single-cell loose-patch recording revealed the sound-evoked burst firing of HB cells as a physiological biomarker of an engram of the learned composite sound. Precise single-cell measurements reveal that it was the same individual HB cells that could invariably embody the learned composite sounds and could reinstate bursting responsivness from dormancy. Despite major differences in timing precision, physiological tagging and IEG tagging methods are not mutually exclusive for identifying memory engrams. In particular, our results that the high-frequency bursting response is a physiological biomarker of an engram could inspire a new path in the future, to explore different IEGs or other molecular markers that exhibit better correlation with bursting neuronal activity[44,45].

A complete identification of HB cells, owing to concurrent technical limitations, requires a highly challenging experimental method such as the combination of two-photon Ca$^{2+}$ imaging and targeted single-cell loose-patch recording. Without a direct verification by targeted loose-patch recording, the HB cells identified by a calibrated two-photon Ca$^{2+}$ imaging should be regarded as 'putative'. To this end, the threshold detection algorithm (note that the detection threshold varies per Ca$^{2+}$ sensor and thus need to be calibrated) is equivalent to the threshold detection of IEG expression level (as reported by the brightness of their relevant fluorescence marker). On the other hand, given the rapid progress of voltage sensors in recent years[46], it may become feasible in the near future to directly identify the sparse HB cells among large populations of neurons in one step through voltage imaging. Regardless of the type of physiological fluorescence sensor being used, the identification of HB cells in-vivo will still require unambiguous single-cell resolution in the opaque mammalian cortex, for example, using multi-photon imaging.

HB cells in AuC L2/3 offer a very fast means of access to memory (latency ~30 ms, see also Supplementary Fig. 1), immediately following the early thalamocortical sensory input activations in the granular layer. Note that our results are not inconsistent with the exisiting knowledge regarding learning-related representation plasticity of tone tuning preferences in the primary auditory cortex (A1), as described in previous studies[47]. However, the nature of HB cells themselves, robustly responding in a high-frequency bursting (at the order of magnitude of ~100 Hz) specifically to a learned composite sounds, may suggest that they serve a unique role in processing and remembering more complex, naturalistic, and behaviorally relevant sound stimuli. Furthermore, beyond the initial observation of HB cells existing in A1 in our previous study[34], the current study shows that HB cells are widely distributed throughout the entire auditory cortex including all classical tonotopic areas[39] (A1, A2, AAF). For each of such learned, behaviorally relevant composite sounds there is a sparsely and widely distributed subpopulation of cells as HB cells in auditory cortex that meet the four theoretical defining criteria of an engram[1] (for a summary, see Fig. 6): thus, these physiologically-defined HB cells are specific single-cell candidates of a cellular engram underlying long-term memory storage in auditory cortex.

## Methods
### Animals (all datasets)
C57BL/6J male mice (56–70 days old) were obtained from the Laboratory Animal Center of the Third Military Medical University. All experimental procedures were performed in accordance with institutional animal welfare guidelines with approval of the Third Military Medical University Animal Care and Use Committee. Mice were socially housed (3–4 mice per cage) under a 12 h light/dark cycle (lights off at 7 pm) and provided with ad libitum feeding.

### Initial surgery and GCaMP6m labeling of AuC L2/3 (all datasets)
Anesthesia of mice was induced with 1.5% isoflurane in pure O$_2$ (0.5 l/min). The pure O$_2$ gas was introduced by a flowmeter into a calibrated commercial vaporizer (R640, RWD Life Science Co., Ltd.) containing isoflurane (100%, 24.5 mM). Mice were kept at a relatively constant anesthetic depth, characterized by a loss of reflexes (such as tail clips) and a respiratory rate of 80–110 breaths per minute. Then mice were placed in a stereotactic frame with a heating pad (37.5 °C). After removing the skin, a small craniotomy (~0.5 mm in diameter) was performed above the dorsal auditory cortex (AuC) at −3.1 mm anteroposterior (AP) and −3.8 mm mediolateral (ML) from bregma. The inclined electrode (diameter: ~20 μm, Angle: 65° to the horizontal) filled with ~80 nl AAV2/9-hSyn-GCAMP6m-XC-WPRE virus was slowly inserted underneath the surface of AuC pia and moved forward 1.4 mm to reach the target layer 2/3 of AuC (~0.3 mm depth from pia) and inject the virus (20 nl per minute). Note this geometrical arrangement of injection path for minimizing tissue damage in the desired central zone of AuC for neuronal functional imaging. The virus was slowly injected while the electrode tip was gradually withdrawn until a depth of 1.2 mm was reached. Then the electrode was held steady for 5 minutes before retracting. After injection, the cranial window was filled with bone wax and the surgical incision was closed with tissue glue (Vetbond, 3 M Animal Care Products). Analgesics (Meloxicam, 1 mg/kg, Boehringer Ingelheim) were applied postoperatively once a day for two days to reduce inflammation. After this initial surgery for GCaMP6m labeling, mice returned to their home cage to recover for ~2 weeks before the next surgery for head-post and cranial window implantation (below).

### Second surgery for head-post and cranial imaging window implantation (all datasets)
Steps 1 and 2 are performed in the same day for mounting a head-post and planting the imaging window.

Step 1: Head-post implantation.
Two weeks after the GCaMP6m labeling step as described above, mice were anesthetized for a second surgery, followed by hair removal,

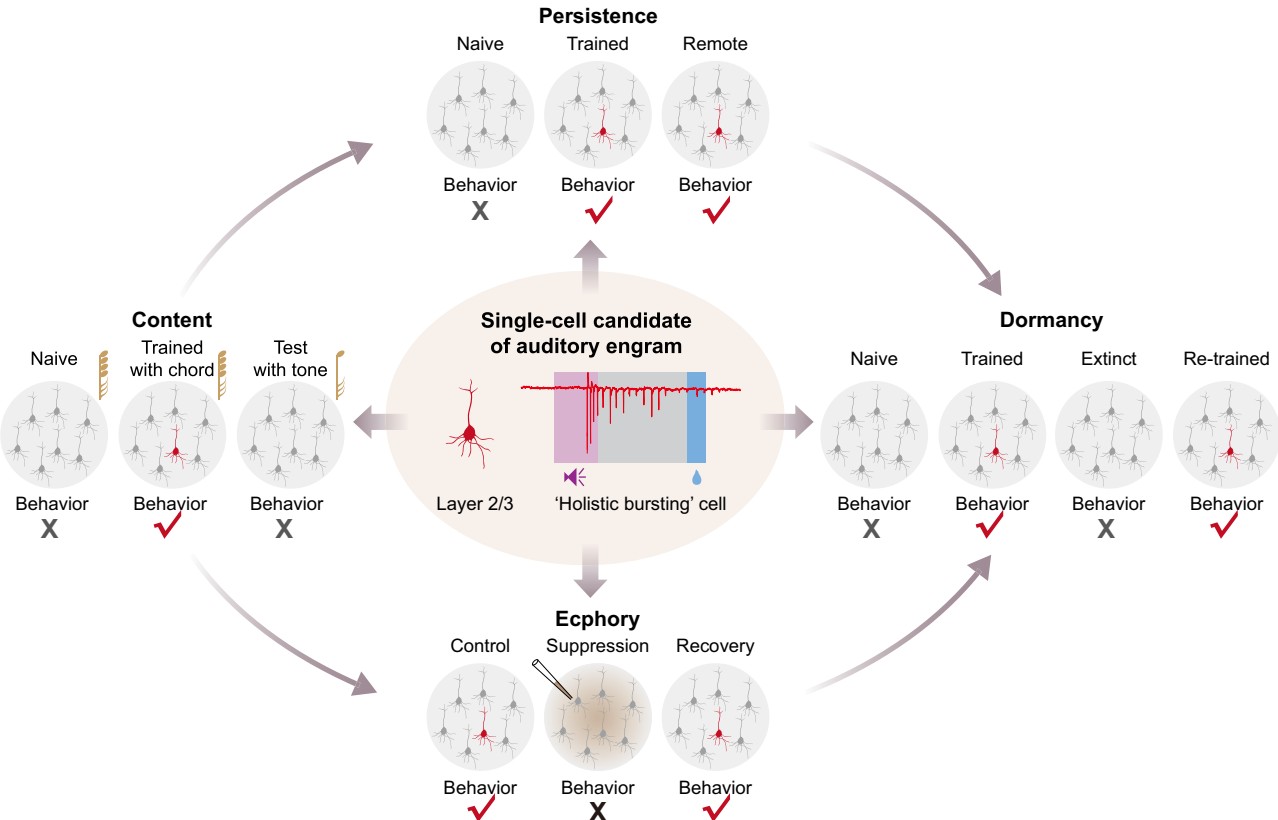

**Fig. 6 | Summary: holistic bursting (HB) cells are single-cell candidates of auditory engram underlying long-term memory storage.** Schematic of the study showing that the 'holistic bursting cell' was investigated on each of the 4 aspects of a full definition of an 'Engram' coined by Richard Wolfgang Semon, namely 'Content', 'Persistence', 'Ecphory' and 'Dormancy'. On the first block (left-side), the musical note of a 4-chord represents the 4-tone-composite chords used during learning, and the musical note of a single tone represents one of the component pure tones used during testing after learning completed (but not during learning). The red-marked cell represents a 'holistic bursting cell' when it exhibits bursting specifically to a learned chord, note that the same cell at the same position can also appear gray as other cells, meaning that they are not bursting in that condition. This summarizes findings from both a previous study[34] ('Content') and this study ('Persistence', 'Ecphory' and 'Dormancy').

skin resection and then clearing of skull fascia with 3% $H_2O_2$ in the head-post footprint area. A custom head-post[34,35] was fixed to the skull with dental cement (Superbond, Sun Medical Co., Ltd.), and the right and bottom edges of the head-post edge were aligned (Right edge: −3.0 ~ −3.6 mm (AP), Bottom edge: −3.3 ~ −3.6 mm (ML)).

Step 2: Cranial window implantation.

A circular cranial window (2.5 mm diameter coverslip) was implanted over the right AuC. To this end, the muscle and skull fascia above the temporal skull were removed after local lidocaine (2%) injection. Afterwards, a custom-made plastic chamber was fixed to the skull with dental cement (Superbond, Sun Medical Co., Ltd.) and a small craniotomy (~2.7 mm in diameter, the center point: AP: −3.0 mm, ML: −4.5 mm) was performed. The dura was removed, and the craniotomy was polished to match the size of the coverslip. A coverslip was carefully placed on top of the cortex with mild compression by tweezers. The coverslip was sealed with UV-curing dental cement (Tetric N-flow, Ivoclar Co., Ltd.). Antibiotics (Cefazolin, 500 mg/kg, North China Pharmaceutical Group Corporation) were administered before surgery, as well as until 3 days after surgery. 1 week of post-surgery recovery was needed before learning and imaging experiments could commence.

**Auditory stimulation (all datasets)**
Sound stimuli were delivered by an ED1 electrostatic speaker driver and a free-field ES1 speaker (Tucker Davis Technologies). The distance from speaker to mouse ear (contralateral to the AuC that was imaged) was ~6 cm. Sound stimuli were produced by a custom-written, LabVIEW-based program and transformed to analog voltage (1 MHz sampling rate, USB-6221 National Instruments). For training with two chords (Figs. 1, 2, 4), Chord 1 consisted of 2.0, 2.7, 3.6, 4.9 kHz, ~78 dB sound pressure level (SPL); Chord 2 consisted of 8.9, 12.1, 16.3, 22.0 kHz, ~71 dB SPL; For training with broadband noise (BBN), we generated a waveform segment of BBN (Bandwidth 0–50 kHz, ~65 dB sound pressure level (SPL)) and used the same waveform segment for all experiments involving BBN (Fig. 3). For testing tone-evoked neuronal responses after the initial training course with chords (Fig. 1), we used 8 pure tones (exactly those 8 tones that each 4 composed one of the two chords): 2.0, 2.7, 3.6, 4.9 kHz (together makes chord 1); 8.9, 12.1, 16.3, 22.0 kHz (together makes chord 2). The loudness of the 4 lower tones were set to ~78 dB SPL to match the loudness of chord 1, and the loudness of the 4 higher tones were set to ~71 dB SPL to match the loudness of chord 2. Note that pure tones were not involved in any phase of the training (initial, reversal, dissociation, and re-association), they were only used for a subset of experiments after the initial training with chords to test the 'holistic' property of the emerged bursting responsive cells (Fig. 1). The background noise level was kept at ~55 dB SPL for all experiments. Neither visible ambient light nor other sensory stimuli were present in any of the training/imaging experiments. All sound stimuli, regardless of which composition, were presented for a duration of 50 ms over 5-6 trials per sound. Note that the short duration of sound stimuli was crucial to the study paradigm as it allowed a temporal segregation between sensory recognition and behavior decision.

### Initial associative training (all datasets)

After 1 week of post-surgery recovery, mice were head-fixed on a custom recording rig for habituation prior to training (~2 h over 2-3 days). No sound stimulus was presented during the habituation. If any signs of stress were displayed, the mice were removed from the recording rig immediately. More habituation days were appended till the head-fixed mice no longer exhibited any visible signs of stress.

Water droplet was dispensed by an automatically controlled water delivery pump (duration, 20 ms, dispensed volume estimated ~5 µl), triggered by the presentation of a sound stimulus (duration: 50 ms, latency: 100 ms from sound stimulus end). The animal's licking behavior was monitored by an infrared camera (frame rate 30 Hz). To avoid the possible effect of rhythmic predicative responses, the inter-trial-interval was randomized in the range of 5–10 s, and the two chords were interlaced in a randomized sequence.

Note that there was no punishment for wrong licking events, and the mice did not show visible signs of discomfort on the recording rig. The water dispensing spouts were positioned at a distance of approximately 3–4 mm from the animal mouth such that the animal had to voluntarily stretch out its tongue to get the water.

The body weight of water-restricted mice was tracked and remained in the range of 80–90% of their body weight before water restriction. Note that the water restriction level was individually adjusted per each animal, such that at the beginning of each day's training & imaging session the animals did not spontaneously lick too much (due to thirst), i.e., spontaneous licking rate <10%.

Day-by-day chronic two-photon Ca²⁺ imaging was performed from the very beginning of the associative training (see below).

### Behavioral data analysis (all figures)

We used custom-written code in MATLAB to extract the tongue movements and determine the timepoint when the tongue touches a water spout (left or right). A trial was defined as a success if the animal's first lick touched the correctly assigned spout within 1000 ms following the sound presentation. Importantly, in case a spontaneous licking event coincidentally started before the end of the sound stimulus, the trial was discarded (but no punishment was given to the animal) and did not contribute to the denominator for the calculation of success rate. The behavioral analysis included all valid trials from the beginning of the session until the moment when altogether 20 droplets of water had been consumed. If the animal did not consume the water reward before the next trial began, a new droplet would replace the existing droplet. Usually in the 2-chord scenario, animals tended to lick on both spouts one after the other, but we only considered the first hit being on the correctly assigned spout as a success trial, and there was no punishment applied to the animal if it licked first on the wrong side.

### Chronic two-photon Ca²⁺ imaging in AuC L2/3 (all datasets)

**Image acquisition.** Two-photon Ca²⁺ imaging was performed using a custom-built two-photon microscope system (LotosScan 1.0, Suzhou Institute of Biomedical Engineering and Technology) with a mode-locked Ti:Sapphire laser working at 920 nm (Mai Tai DeepSee, Spectra Physics) and 40x/0.8 NA water-immersion objective (Nikon), field-of-view (FOV) size 300 × 300 µm, 20 frames/second.

**Chronic imaging.** Continuous two-photon Ca²⁺ imaging was performed daily over a period of 10–16 days. To minimize the bleaching of cells, we used low laser illumination (~30 mW), so that the bleaching of the GCaMP6m sensor was negligible over the continuous imaging period. All FOVs were at a depth of 110-300 µm below the pial surface and 5-6 FOVs were collected per animal over multiple days of imaging. The repeated imaging FOVs were identified on consecutive days based on superficial blood vessels and nearby blood vessels, then further refined by visually matching reference images acquired from precedent days. Note that by the end of the chronic imaging the overall consistency was checked once again, and only those FOVs that consistently matched themselves day-by-day qualified for the analysis.

### Targeted single-cell loose-patch recordings in behaving mice and definition of burst events (Fig. 1)

A 'shadow-patching' procedure was used in loose-patch recording experiments of behaving mice. After continuous imaging of the targeted cells over days, a borosilicate glass microelectrode with a tip resistance of 5–7 MΩ was filled with normal ACSF and 30 µM Alexa-594. The Alexa-594 enabled visualization of the electrode in the two-photon microscope. The electrode was mildly navigated through the cortex from an insertion angle of ~35 deg to the depth range of L2/3. Loose-patch configuration was formed with cells that are identified by GCaMP6 imaging. This procedure was also also used for local application of isoflurane in AuC L2/3. The electrophysiological recordings obtained from targeted cells with an EPC10 amplifier (HEKA Elektronik). Electrophysiological data were filtered at 10 kHz and sampled at 20 kHz using the Pulse software (HEKA Elektronik). Electrophysiological and Ca²⁺ imaging data were analyzed off-line by using Igor Pro 6.0 (Wavemetrics). A burst event (loose-patch recording) is defined as a multi-spike event including ≥3 spikes consecutively with an inter-spike-interval ≤50 ms each. Both sound-evoked and spontaneous activities were recorded in multiple sessions of 30–50 s duration, the only difference being that for sound-evoked activities the sound stimuli (either chord or tone, in randomized sequence) were presented with randomized inter-stimulus-intervals and for spontaneous activities there was no sound stimulus presented. The spontaneous burst events rate for a cell was determined as the total number of spontaneous burst events divided by the total recording time of that cell.

### Left-right equal-valued reversal training (Fig. 2)

After the first 6 days of initial training, from the beginning of training session of day 7 on, the sound-water association assignments were reversed between left and right sides without any adaptation period. From the perspective of the animals, in the initial training course the association assignments were chord 1 to right spout and chord 2 to left spout, in the reversal training, the assignments were chord 1 to left spout and chord 2 to right spout. In both the initial and the reverse training course, the two water spouts dispensed equal amounts of water per stimulus event in order to maintain equal value for each sound.

### Dissociation and re-association training (Fig. 3)

The 'BBN' training scenario is almost identical to the '2 chords' training scenario (Figs. 1 and 2) except that there was only one sound stimulus (broad-brand-noise synthesized with a 'frozen' waveform that remains constant from trial to trial) and one water dispenser. The learning speed is faster in this scenario as there is only one association to be established instead of two. Mice first reached high behavioral performance levels (>80%) for at least two days (day 4 and day 5). From day 6 on, in the dissociation phase, the sound-water associations were interrupted by withholding water delivery following the sound stimulus. Mice were provided with *ad libitum* water after the end of each session but without any sound stimulus. The dissociation phase took 4 days (day 6, 7, 8, 9). Then, the same mice experienced re-training phase, whereas the sound stimulus was associated with water delivery again for another 4 days (day 10, 11, 12, 13). Note: data for this figure are pooled from two different datasets, one was a pilot dataset which included 5 animals with only 4 days of imaging (day 1, 5, 9, 13) and the other was a more complete dataset which included 10 animals with imaging in every day of the whole course of 13 days. For the analysis in Fig. 3b, c only the latter dataset was used, and for Fig. 3d both datasets were pooled together.

## Wide-field epifluorescence imaging (Fig. 4)

Wide-field imaging was performed through a glass coverslip (3 mm diameter) to the auditory cortex by using a 470 nm LED light source. A 4X, 0.1 NA objective (Olympus) was used for imaging and the emission light was collected and imaged by a CMOS camera (ximea MC089MG-SY) at 10 Hz. All the cortical maps (1088 ×1088 pixels) were captured by using consecutively 10 repeated stimuli (50 ms duration, 5 s inter-trial interval). Four pure tones (4, 8, 16, and 32 kHz) and the two chords (same as used for learning experiments) were tested sequentially. Then the cortical maps were downsampled to 544 ×544 pixels for further analysis. A response zone for a certain tested stimulus was calculated as follows: first obtain the $Ca^{2+}$ response per each pixel as $\Delta f/f = (f\text{-}f0)/f0$, with the baseline fluorescence (f0) and response fluorescence (f) obtained by averaging images 1000 ms before and 2000 ms after the sound stimuli, respectively. Then apply a spatial Gaussian filter (kernel size 5×5 pixels) for smoothing. The $Ca^{2+}$ response peak in the sound evoked response period was defined as the frame with the maximum signal change within a 0.5 s period following stimulus onset. The size of the response area was estimated from the number of pixels in which $\Delta f/f$ was larger than half maximum amplitude.

For normalizing brain regions of all mice, the geometric centroids of each brain region were calculated to estimate affine matrices according to the matching relation of centroid points. Then the alignment among those brain regions was achieved by registration of an affine transformation.

## Local application of isoflurane in AuC L2/3 (Fig. 5)

For local isoflurane application experiments, we exposed the right AuC of trained mice (initial training with chords, day 1 to day 6, GCaMP6m-labeled) as described above, by means of an acute surgery of craniotomy for imaging. Similar to the surgery procedure of mounting a chronic imaging window, the animal was anesthetized by gaseous isoflurane and kept on a heating pad (37.5 °C). Afterwards, a custom-made plastic chamber was fixed to the skull with dental cement (Superbond, Sun Medical Co., Ltd.) and a small craniotomy (~2.7 mm in diameter, the center point: AP: −3.0 mm, ML: −4.5 mm) was performed. The dura was left intact under the craniotomy in order to minimize the tissue damage during acute surgery. After the surgery, the gaseous isoflurane in the inhalant was removed, and the animal was transferred to the recording rig. The craniotomy was filled with 1.5% low-melting-point agarose. The plastic chamber was perfused with artificial cerebral spinal fluid (ACSF) containing in mM: 125 NaCl, 4.5 KCl, 2 $CaCl_2$, 1 $MgCl_2$, 26 $NaHCO_3$, 1.25 $NaH_2PO_4$ and 20 glucose (pH = 7.4). The agarose application and ACSF perfusion were derived from experiences for the protection of brain tissue after the acute surgery with an opened craniotomy in the same day. The two-photon neuronal imaging and isoflurane intervention test were performed from >1 h after animal transferring. Note that this surgery with gaseous isoflurane as general anesthetics was retrospectively justified with experimental data showing that both the learned sound-associated licking and the neuronal responses recovered in ~1 h after general anesthesia.

In the unilateral local isoflurane application test, aqueous isoflurane solution was injected via a micropipette inserted into AuC L2/3 guided by two-photon microscopy. The isoflurane was first dissolved in DMSO with 20% F127, then further diluted in ACSF to achieve the desired volume fraction (2% for data shown in Fig. 5, 1% for the dosage test in Supplementary Fig. 4) in the injection solution. The DMSO as intermediate solvent was required because isoflurane is hydrophobic. To dissolve isoflurane, mild shaking of the mixing tube was performed by hand instead of using a vortex mixer machine. Alexa-594 (30 μM) was also included in the injection solution for visualizing the injection under two-photon microscope. Note that all these adjuvants were calculated and mixed with a stock ACSF solution with adjusted concentrations of the ionic ingredients, such that the final injection solution had the same ion concentrations and osmolarity as of the normal ACSF. Same as in the "shadow patching" procedure, a borosilicate glass micropipette (tip resistance of 4−7 MΩ) loaded with the injection solution was mildly approaching the targeted focal plane in right AuC. While the animal was awake and behaving, a single bolus of injection was applied for 10−20 s with 200−300 mbar pressure. We performed the behavior test and two-photon imaging in several time blocks before and after the injection and defined the stages accordingly ('before': ≥20 min before injection onset; 'isoflurane': 1−20 min from injection end; 'recovery': ≥60 min after injection end).

In the bilateral local isoflurane application test, isoflurane solution (2% volume concentration) was bilaterally delivered to L2/3 of AuC by two glass micropipettes each inserted into one side (no imaging performed, only behavioral monitoring). The injection protocol utilized a similar version of the AAV injection as mentioned above, a small craniotomy (~0.5 mm in diameter) was made above AuC at −3.1 mm AP and 3.8 mm ML from bregma. Two glass micropipettes, each with a tip diameter of ~20 μm, were inserted via the bilateral craniotomy. An estimated volume of ~80 nl was injected into each side of AuC in behaving mice.

## Quantification and statistical analysis (all datasets)

Data analysis was performed with a custom-written analysis software pipeline using LabVIEW 2017 (National Instruments), Igor Pro 6.0 (Wavemetrics), Image J (NIH), Prism 8.4 (GraphPad), SPSS22 and Matlab 2018b (Mathworks). Source codes are included in an online zip file and also available upon request.

**Image registration.** To perform batch calibration for brain motion along the imaged focal plane (x-y motion), which was caused by movements during licking behaviors, we used a custom correction software (Matlab 2018b, Mathworks). The correction codes were written based on the image alignment software TurboReg (ImageJ, NIH). We performed frame-by-frame alignment for the imaging data with a translation algorithm, where within each imaging day, the imaging data was registered to the average image (the first 100 frames).

**Basic imaging data analysis.** For extracting fluorescence signals, we visually identified neurons and performed the drawing of regions of interests (ROIs) based on fluorescence intensity. To avoid using overlapping pixels resulting by cell masks, the top 80% of pixel weights were included into the ROI and any remaining pixels identified in multiple cell masks were excluded. For data from continuous $Ca^{2+}$ imaging, we tracked the same FOV based on the last training day. We removed the outer ~10% of the image from each ROI to account for edge effects or imaging deviation. The $Ca^{2+}$ activity changes (f) were extracted using a custom implementation of common methods[34,35]. The fluorescence changes (f) were calculated to correspond to pixel values in each specified ROI. Relative fluorescence changes ($\Delta f/f$) were used in most $Ca^{2+}$ response data in this paper. $\Delta f/f = (f - f_0)/f_0$ were calculated as $Ca^{2+}$ signals, where the baseline fluorescence $f_0$ was estimated as the 25th percentile of the entire fluorescence recording. Note that all analyses of neuronal $Ca^{2+}$ signals in all conditions (including those under isoflurane intervention) are restricted to sound-evoked responses only, accepting the first peak within 500 ms from sound stimulus onset.

## Statistical tests

For simple rate calculations of samples from the same group, we used an online calculator for chi-square test to compare two rates of the same denominator (https://www.medcalc.org/calc/rate_comparison.php).

For comparing data (i.e., behavior and amplitude) from two different sample groups was used Two-sided Mann−Whitney test (unpaired) or two sided t-test (paired) to test whether or not there is a significant difference between them. Group comparisons (in different

conditions) were made using one way analysis of variance (ANOVA) followed by Tukey post-test.

All physiological data ($Ca^{2+}$ or loose-patch responses) of a defined sample group are presented as median\25–75% IQR, and all behavior data with error bars are presented as mean ± sem. Most data presented in figures with box-and-whisker plots indicate median (center line), 25–75% IQR (box), minimum and maximum (whiskers), and behavior data with error bars are presented as mean ± sem.

In addition, we also applied the Bayes factor hypothesis testing[42] on the isoflurane intervention dataset (Fig. 4). We used the two-sided Bayes factor $BF_{10}$ and a criterium value $x = 3$ to define the outcome, as follows: $BF_{10} \leq 1/3$: evidence of absence (favoring null hypothesis $H_0$, no difference between groups); $1/3 < BF_{10} \leq 3$: absence of evidence (data insufficient to support alterative hypothesis $H_1$); $BF_{10} > 3$: evidence of presence (favoring $H_1$, corresponding to $P < 0.05$ in standard tests); $BF_{10} > 10$: strong evidence of presence (strongly favoring $H_1$, corresponding to $P < 0.01$ in standard tests).

### Reporting summary
Further information on research design is available in the Nature Portfolio Reporting Summary linked to this article.

## Data availability
Large chronic raw image data from wide-field imaging and two-photon imaging are available upon request from the corresponding author. Source data underlying Figs. 1–5 and Supplementary Figs. 1–4 are available as a Source data file. No datasets that require mandatory deposition into a public database were generated during the current study. Source data are provided with this paper.

## Code availability
Customized software codes are available at Github (https://github.com/hbcell/Code). More software codes supporting this study's findings are available from the corresponding author upon request.

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

## Acknowledgements

The authors would like to thank Dr. Israel Nelken for introducing the specific composite chord stimuli in the behavior learning paradigm, Dr. Rodrigo Quian Quiroga for inspiring the equal-valued choice reversal experiment, and Jia Lou for cartoon art and figure layout. We also thank Weiyi Liu, Xiaojing Tang and Zhehao Xu for implementing and maintaining software codes for image analysis. This study is supported by grants from National Natural Science Foundation of China (31925018, 32127801), the Jiangsu Provincial Big Science Facility Initiative (BM2022010), the Scientific Instrument Developing Project of the Chinese Academy of Sciences (No. YJKYYQ20200052), the Basic Research Pilot Project of Suzhou City (SJC2021021), the Jiangsu Provincial Innovation and Entrepreneurship Team Fund, the Deutsche Forschungsgemeinschaft (DFG, German Research Foundation) – Project-ID 425899996-SFB 1346 (to S.R. and J.P.), the Saxony-Anhalt Excellence Initiative pilot project "Cognitive Vitality". X.C. is a fellow of the CAS Center for Excellence in Brain Science and Intelligence Technology.

## Author contributions

X.C. and H.J. designed the study. R.L. primarily performed all the in-vivo experiments and data analysis with support from the other authors as follows: custom-built instruments and technical support: J.L., Zhenqiao.Z., M.L., W.J., H.L., Zhikai.Z., S.W., L.L., Y.T., H.J.; establishing experimental methods: J.H. (particularly the day-by-day chronic imaging and targeted loose-patch following chronic imaging), L.L., Zhikai.Z. (particularly the combination of wide-field imaging and two-photon imaging), Susu.L., M.W., H.C., J.Z., X.C.; establishing data analysis methods: Shanshan.L., X.L., X.C., H.J.; data inspection: D.H., S.R., J.M.P.P., X.C., H.J.; result interpretation and manuscript writing: J.M.P.P., X.C., H.J.

## Funding

## Competing interests

The authors declare no competing interests.
