## [Peer Review File · Nature Communications]

Holistic Bursting Cells Store Long-term Memory in Auditory CortexREVIEWER COMMENTS

Reviewer #1 (Remarks to the Author):

In the manuscript “Holistic Bursting Cells Store Long-term Memory in Auditory Cortex,” Li et al. demonstrate the existence of “holistic bursting” (HB) cells in the auditory cortex after an associative auditory learning task, while employing 2-photon calcium imaging during learning. The behavioral task consists of associating 2 different chords to left or right lick spouts, to obtain a water reward. The authors demonstrate that mice learn this task rather quickly, within 6 days (Fig. 1), that the memory of the task is persistent long-term (Fig. 2), and can be quickly relearned within 4 days after reversal of left-right contingencies (Fig. 2). The authors use in vivo loose patch recordings to verify the presence of HB cells (Fig. 1c-d). Most noteworthy, the authors demonstrate that HB cells emerge from quiescence during the period of learning (Fig. 1); HB cells are specifically associated with the learned chords and not behavioral choice (Fig. 2); and HB cells return to dormancy and reemerge with dissociation and re-association, respectively. Moreover, the authors claim that these HB cells are causally responsible for the memory of the behavioral task because application of local isoflurane, which selectively inhibits bursting cells, impairs task performance (Fig. 4). Overall, the findings in this manuscript move the field forward by establishing a model of memory “engram” cells in sensory cortex, in a real-time imaging setup during learning. As such, these methods both shed new light on possible mechanisms of memory in sensory cortex, attributed to HB cells, and develop techniques for studying these mechanisms, building upon previous literature on immediate early gene expression. Overall, the methods and findings are relatively strong; however, there are some questions and comments that should be addressed, outlined below:

Major Comments:

1. If my understanding is correct (also see minor comment #3), the authors use loose patching of 4 HB cells, shown in Fig. 1d, to demonstrate and define the properties of HB cells. Namely that they selectively respond to a chord, and the response is greater than the sum of the constituent tones. Then, they use the calibration curve obtained from these cells to establish a threshold $dF/F > 0.8$, which they apply to the entire imaged population to identify all HB cells. By my count, they find a total of 105 “HB” cells (Figs 1-2b $n=43$; Fig 2c $n=12$; Fig 3 $n=32$; Fig 4 $n=19$). By labeling these cells “HB cells”, the authors’ logic necessitates that all 105 cells have the same firing properties as the 4 cells sampled via loose patching, which is an assumption rather than a definitive conclusion. I understand that it is not feasible to loose patch every cell that exhibits $dF/F > 0.8$; however, one suggestion is to explicitly state that the HB cells identified solely via imaging are “putative” (not definitively) HB cells. Additionally, a discussion of this caveat should be added to the Discussion section.
2. The interpretation of results in Fig. 2 also seems like a logical stretch, and warrants further discussion of alternative explanations. The authors conclude that the HB cells “embodied the learned sounds but not the behavioral choices” (line 140), because they observed zero cells that switched from one chord to the other during reversal training. I agree that for the 29 cells with persistent responses to the same

chord through reversal training, these 29 cells embody the learned sounds but not behavioral choice. However, they also observed 25 cells that either lost or gained responsiveness through reversal training. One explanation for these 29 cells could be that their responses are at least partially dictated by behavioral choice: for example, a HB cell may selectively respond to chord 2, but only with a left reward.

3. For Fig. 3b-c and 3d, the numbers of cells and mice do not match. These figures are presumably showing data from the same experiments, but N=10 mice in 3b-c, and N=15 mice in 3d (figure legend). Moreover, the text describing Fig. 3 states “894 imaged cells pooled from 15 mice”. Fig. 3c legend states 895 cells, but only 10 mice. Is this simply a typo or clerical error? Or were some mice included in 3d but excluded from 3c? If the latter, the authors should explain why some data were excluded from analysis.

4. What is the rationale for switching to broadband noise in Fig. 3? Why not present 1 chord (rather than 2), which would have been more consistent with the other experiments? Why use a different stimulus?

Minor Comments:

1. In general throughout the manuscript, the authors should avoid using vague terms such as “some” or “most”, and provide specific values. For example: Results line 80 states “For some animals...”. It is more helpful to readers to simply state the fraction of mice. E.g. “For 3 out of 13 animals...”

2. Results line 84, and Extended Data Fig. 1: How many cells were included in the data for the calibration curve? N=4 loose-patched cells from 3 mice?

3. As written, when the authors first describe “cells tagged in the last day of the training course” (line 100), it is slightly unclear whether they are referring to cells that were loose-patched on day 6, or all of the cells identified as HB based on Ca fluorescence. This is confusing because this section immediately follows the section describing loose patching on day 6, so “cells tagged on day 6” at first seemed to refer to these cells. However, upon further re-reading, it became clear that throughout the manuscript, the HB cells “tagged” on day 6 include all cells that exhibited bursting evidenced via fluorescence imaging, based on the calibration criteria. If I understand correctly, only the 4 cells shown in Fig. 1d were loose-patched. It would be helpful if the authors can further clarify – perhaps by adding a clearer statement(s) that the loose patch technique was used in a subset of cells to verify holistic bursting properties, then the calibration curve criteria was applied to all neurons to “tag” putative HB cells in the imaged population.

4. Fig. 1b legend: Scale bar distance is not provided

5. Methods: how many trials were obtained during the imaging session(s)? How many trials were averaged to get the average response for each cell?

6. Related to comment 5: Methods line 547 states that analysis included trials from a session until “20 droplets” of water were consumed. Therefore, when mice are performing at ~100% performance, this implies that sessions only lasted ~20 trials, and would be a rather short testing/imaging session. Please clarify.

7. Methods line 566: Are the “5-6 FOVs per animal” per session (for a total of 30-36 FOVs for 6 days)? Or 5-6 total over the multiple days of imaging? Please clarify.

8. Methods line 601: Is the procedure described here for pipette insertion/ACSF the same as what is used for loose patch recordings (line 571). If so, these details should be provided in the section on loose patch recordings as well.

9. Methods line 678: Why do the authors use Bayes factor testing for the data in Fig. 4? What is the rationale for choosing this approach here?

Reviewer #2 (Remarks to the Author):

The manuscript from Li et al. entitled "Holistic Bursting Cells Store Long-term Memory in Auditory Cortex" described results from a series of experiments to investigate how auditory cortical neurons establish and store behavioral relevant sound information during learning. Chronic 2-p Ca²⁺ imaging was used to reveal emerged holistic bursting cells (HBCs) in layer 2/3 of mice auditory cortex during an auditory associative learning. Reversal and dissociation of the learning, together with the isoflurane-mediated silencing of these neurons dampened task performance supported the causal effect of HBCs in the auditory associative memory.

The experiments were logically well-designed, and a series of sophisticated techniques were employed. Results are convincing and the finding is significant. I have some minor concerns detailed below:

1. The recordings were performed focusing on the layer 2/3 of the auditory cortex. It does not rule out the possibility that other neurons in deeper layers are also involved in the learning and memory storage. Isoflurane manipulation may (most likely) also silence neurons in the deeper layers as well. Discussion of this possibility is needed and conclusions need to be precise accordingly.

2. With the similar notation, the claim of "...HB cells as long-term memory storage in the sensory neocortex." is too broad. The result in this manuscript is only supporting the conclusion for the auditory cortex.

3. In Figure 4 the analyses of Ca²⁺ signals showed that isoflurane delivery significantly silenced HBCs but not for other cells. However, since isoflurane is believed to act on the neurons via the same mechanism, it is highly possible that this manipulation is a general suppression of the auditory cortex (including the above threshold and subthreshold activities may or may not detected by the Ca²⁺ imaging analyses). This is important to reach the causality conclusion of the HBCs in this associative memory. This limitation should be discussed.

4. It is not mentioned in the text or figures about the GCaMP6m vector. Which promoter was used? Are both excitatory and inhibitory neurons labeled and imaged? Are they separated in the analyses?

5. It is not clear why in figure 3 a different task was used to assess the dissociation effect. This is an auditory detection task and the other used in Figs 1/2/4 is an auditory discrimination task. The auditory cortical neurons may or may not function in the same way on these tasks. The limitation should be discussed.

Reviewer #3 (Remarks to the Author):

The manuscript by Li et al. is exciting and timely conceptually, technically impressive, and appropriately grounded in a deep and rich literature in the neurobiology of memory. The authors report a physiological signature of auditory associative learning in the auditory cortex, specifically in "bursting" responses of select layer 2/3 neurons (though presumably this kind of bursting activity is possible in other layers, but undetected using the approach in the current report). Strikingly, several tests of the nature of the putative physiological signature of learning show characteristics that qualify these burst responses as an "engram" of the salience of a specific sound chord that is paired with water reward. The bursting cells emerge with learning, survive reversed behavioral contingencies (likely due to the salience of the sounds being unchanged), are long-lasting (up to a month), track extinction by reducing in response activity with reduced sound salience, and (at least in part, some of the cells) reinstate bursting once the chords are again made behaviorally salient.

This is a remarkable set of findings that open the door to many novel questions that the authors may address: For example, what is the circuit function of these layer 2/3 cells? In other words, what cells are downstream of this activity? Upstream? Is there any predictable topography of these cells based on, say, the acoustic frequencies of the salient chords? At a molecular level, novel questions emerge for the single-cell: What are the cellular consequences of burst activity with respect to gene expression, calcium-related signalling cascades, etc., that could predict persistent changes to cellular functions that support bursting activities?

The authors go on to perform a causal experiment using a clever reversible manipulation using isoflurane to decrease bursting activity without obliterating responsivity more generally across the auditory cortex. Remarkably, the selective decrease in bursting blunts the behavioral performance of the task, which returns to high performance levels after wash-out. While it is difficult to interpret the specificity of the change in behavior to only this task and chord stimuli, the result is nonetheless compelling as a test of "necessity" of these cells' bursting activity to maintain the behavior, particularly in the context of the other behavioral manipulations reported in the manuscript. Though beyond the scope of the present report, it would be interesting to hear the authors speculate on whether or not bursting activity is "sufficient" to drive cued behavior. Necessity and sufficiency of these layer 2/3 "bursting" responses would definitively identify them as engram cells in the Semon (1921) definition.

On the matter of sufficiency: To what extent do layer 2/3 cells burst spontaneously? Do the authors have inter-trial data that could address whether bursting is only sound-evoked?

On a separate note, I did not see the performance calculation(s) used to quantify behavior across the different tasks and task phases in the methods or figure legends. These are important to include to appropriately interpret learning (acquisition), memory, and task performance abilities.

Reviewer #4 (Remarks to the Author):

In this manuscript, the authors report that neurons in the auditory cortex exhibit plasticity changes after sound cue associative learning. Although I am quite interested in the topic of the study, the results and claims of the study are quite confusing. The study has substantial overlaps of results with the previous publication from the same group titled "Single-neuron representation of learned complex sounds in the auditory cortex (Wang et al, nature commun. 2020)". In addition, the claimed mechanism for "long-term memory storage", or the claimed holistic responses are not supported by the experimental evidence.

1. Previous studies have shown that classical and operant conditioning can lead to a global shift in the tonotopic map toward the frequency of the conditioned sound in the auditory cortex, and that auditory training can alter the tuning properties of individual neurons in the auditory cortex. It's well established that learning can lead to changes in the neural representation of sound in the auditory cortex. The authors should consider these findings when interpreting their results.
2. The recorded auditory cortical responses seem weird. Responses to pure tones are not observed or extremely weak even at around 80dB. If the majority of the neurons did not exhibit robust responses, their comparison with and without isoflurane would not be meaningful. The authors should consider the possibility that the tone used may be outside the receptive field of the recorded neurons, rather than the neurons "emerging from quiescence."
3. The specificity of the persistent plasticity in the selected population has not been convincingly demonstrated, as other recorded cell types have not been included in the comparison. It is essential to address whether other cells in the auditory cortex might also exhibit plasticity after sound-related learning, which seems not consistent with previous experiment results.
4. The loss-of-function experiment presented in the manuscript is not convincing. What's the foundation of the specificity that systematic and local application of isoflurane specifically silenced the observed cells while sparing all other cell types, layers, or regions.
5. The non-specific silencing of the auditory cortex resulting in a deficit of auditory-associated learned behavior should not be surprising. This may not be considered as a supporting evidence for long-term memory storage.
6. What's the basis for the so called "holistic" response? The observation could be explained by the integration function of the recorded neurons being non-linear (e.g., exponential) rather than truly "holistic."
7. The observed responses could be the responses induced by conditioned reward rather than the holistic response to complex sound. The responses correlated better with reward/behavior performance, rather than the complex sound per se, and the response decreases with decrease in behavior performance, and vice versa. Once trained without water reward, the response was largely dismissed.

REVIEWER COMMENTS

Reviewer #1 (Remarks to the Author):

In the manuscript "Holistic Bursting Cells Store Long-term Memory in Auditory Cortex," Li et al. demonstrate the existence of "holistic bursting" (HB) cells in the auditory cortex after an associative auditory learning task, while employing 2-photon calcium imaging during learning. The behavioral task consists of associating 2 different chords to left or right lick spouts, to obtain a water reward. The authors demonstrate that mice learn this task rather quickly, within 6 days (Fig. 1), that the memory of the task is persistent long-term (Fig. 2), and can be quickly relearned within 4 days after reversal of left-right contingencies (Fig. 2). The authors use in vivo loose patch recordings to verify the presence of HB cells (Fig. 1c-d). Most noteworthy, the authors demonstrate that HB cells emerge from quiescence during the period of learning (Fig. 1); HB cells are specifically associated with the learned chords and not behavioral choice (Fig. 2); and HB cells return to dormancy and reemerge with dissociation and re-association, respectively. Moreover, the authors claim that these HB cells are causally responsible for the memory of the behavioral task because application of local isoflurane, which selectively inhibits bursting cells, impairs task performance (Fig. 4). Overall, the findings in this manuscript move the field forward by establishing a model of memory "engram" cells in sensory cortex, in a real-time imaging setup during learning. As such, these methods both shed new light on possible mechanisms of memory in sensory cortex, attributed to HB cells, and develop techniques for studying these mechanisms, building upon previous literature on immediate early gene expression. Overall, the methods and findings are relatively strong; however, there are some questions and comments that should be addressed, outlined below:

We thank the reviewer for their appreciation of our work and the constructive suggestions. We believe that these revisions, as detailed below, have improved the manuscript.

Major Comments:

1. If my understanding is correct (also see minor comment #3), the authors use loose patching of 4 HB cells, shown in Fig. 1d, to demonstrate and define the properties of HB cells. Namely that they selectively respond to a chord, and the response is greater than the sum of the constituent tones. Then, they use the calibration curve obtained from these cells to establish a threshold $dF/F > 0.8$, which they apply to the entire imaged population to identify all HB cells. By my count, they find a total of 105 "HB" cells (Figs 1-2b n=43; Fig 2c n=12; Fig 3 n=32; Fig 4 n=19). By labeling these cells "HB cells", the authors' logic necessitates that all 105 cells have the same firing properties as the 4 cells sampled via loose patching, which is an assumption rather than a definitive conclusion. I understand that it is not feasible to loose patch every cell that

exhibits $dF/F > 0.8$; however, one suggestion is to explicitly state that the HB cells identified solely via imaging are “putative” (not definitively) HB cells. Additionally, a discussion of this caveat should be added to the Discussion section.

In conjunction with the advice of Reviewer #3, we have performed a new set of targeted loose-patch recordings for the revised manuscript. The new data are appended to Fig. 1 and Suppl. Fig. 1. We also included a new discussion section for the identification of HB cells and the caveat of our approach as was suggested.

2. The interpretation of results in Fig. 2 also seems like a logical stretch, and warrants further discussion of alternative explanations. The authors conclude that the HB cells “embodied the learned sounds but not the behavioral choices” (line 140), because they observed zero cells that switched from one chord to the other during reversal training. I agree that for the 29 cells with persistent responses to the same chord through reversal training, these 29 cells embody the learned sounds but not behavioral choice. However, they also observed 25 cells that either lost or gained responsiveness through reversal training. One explanation for these 25 cells could be that their responses are at least partially dictated by behavioral choice: for example, a HB cell may selectively respond to chord 2, but only with a left reward.

We agree that this could be an alternative explanation and we have reworded the discussion section in the revised manuscript, appended a new discussion (right after the data presentation in Fig.2) for this alternative interpretation, and we retained the consensus interpretation for the 29 cells with invariant responses to the same chord through reversal training.

3. For Fig. 3b-c and 3d, the numbers of cells and mice do not match. These figures are presumably showing data from the same experiments, but $N=10$ mice in 3b-c, and $N=15$ mice in 3d (figure legend). Moreover, the text describing Fig. 3 states “894 imaged cells pooled from 15 mice”. Fig. 3c legend states 895 cells, but only 10 mice. Is this simply a typo or clerical error? Or were some mice included in 3d but excluded from 3c? If the latter, the authors should explain why some data were excluded from analysis.

No data was excluded, but Fig. 3d involved additional data obtained from a different cohort of 5 animals which had been pooled from an unpublished pilot dataset (which includes chronic imaging at the 4 observation timepoints but not every day, as in Fig. 3b and 3c). It was indeed an error to report 894 cells for the 10 animals in Fig. 3c, which should have been 755. We apologize for the confusion, and we have corrected the figure legend and main text accordingly.

4. What is the rationale for switching to broadband noise in Fig. 3? Why not present 1 chord (rather than 2), which would have been more consistent with the other experiments? Why use a different stimulus?

In principle, any defined composite sound of a constant waveform can be used for this experiment, they don't have to be one of the two chords used for the other experiments. Note also the BBN used here was also a specific composite sound by composing a broad band of pure tones with randomized phases and amplitudes and its waveform was 'frozen' for all animals and all trials, i.e., not a different noise from trial to trial, but always the same waveform per each trial. Since in the previous study (Wang et al, NC 2020) we have already performed the chronic imaging experiments through a learning cycle till the third phase (dissociation), here in this study we carried on with the same experimental design (using the same 'frozen' BBN) and added a re-association phase, for which, 5 mice were imaged in a pilot experiment that was later pooled together with a new round of experiments (10 mice) for Fig.3d (altogether 15 mice, please also see point 3 above).

Minor Comments:

1. In general throughout the manuscript, the authors should avoid using vague terms such as "some" or "most", and provide specific values. For example: Results line 80 states "For some animals...". It is more helpful to readers to simply state the fraction of mice. E.g. "For 3 out of 13 animals..."

We have removed a number of ambiguous and unclear terms, and accurate numbers are provided in the revised manuscript whenever possible.

2. Results line 84, and Extended Data Fig. 1: How many cells were included in the data for the calibration curve? N=4 loose-patched cells from 3 mice?

Both Fig. 1 and Extended Data Fig.1 have been updated with a much larger dataset of loose-patch recording, after pooling with existing data, the calibration curve now includes data recorded in 14 HB cells and 3 non-HB cells from 12 animals.

3. As written, when the authors first describe "cells tagged in the last day of the training course" (line 100), it is slightly unclear whether they are referring to cells that were loose-patched on day 6, or all of the cells identified as HB based on Ca fluorescence. This is confusing because this section immediately follows the section describing loose patching on day 6, so "cells tagged on day 6" at first seemed to refer to these cells. However, upon further re-reading, it became clear that throughout the manuscript, the HB cells "tagged" on day 6 include all cells that exhibited bursting evidenced via fluorescence imaging, based on the calibration criteria. If I understand correctly, only the 4 cells shown in Fig. 1d were loose-patched. It would be helpful if the authors can further clarify – perhaps by adding a clearer statement(s) that the loose patch technique was used in a subset of cells to verify holistic bursting properties, then the calibration curve criteria was applied to all neurons to "tag" putative HB cells in the imaged population.

Thank you for the suggestion. We revised the text to clarify that the loose-patch technique was used in a subset of cells to verify holistic bursting properties. We also appended a new discussion section regarding the putative HB cells in the imaged population.

4. Fig. 1b legend: Scale bar distance is not provided

Done.

5. Methods: how many trials were obtained during the imaging session(s)? How many trials were averaged to get the average response for each cell?

During each imaging FOV, 5-6 trials per sound (chord 1, chord 2 or BBN) were presented. The identification of a bursting responsive cell (bulk offline analysis) requires at least 3 consecutive responses of $df/f > 0.8$ and these first 3 responses were averaged for that cell. This information has been clarified in the methods section.

6. Related to comment 5: Methods line 547 states that analysis included trials from a session until "20 droplets" of water were consumed. Therefore, when mice are performing at ~100% performance, this implies that sessions only lasted ~20 trials, and would be a rather short testing/imaging session. Please clarify.

The rationale for setting a limit for the analysis window of behavioral performance has been established in the previous study (Wang et al, NC 2020), that the animal could gradually lose motivation to lick once after consuming sufficient amount of water, and we didn't apply any punishment to the animals for not behaving (e.g., a time out period, etc.). Yet, the sound-evoked bursting response (putative, calibrated Ca^{2+} imaging) was highly reliable trial-by-trial, thus we could consecutively perform long sessions of imaging in multiple FOVs on each experimental day.

7. Methods line 566: Are the "5-6 FOVs per animal" per session (for a total of 30-36 FOVs for 6 days)? Or 5-6 total over the multiple days of imaging? Please clarify.

The latter one, 5-6 total FOVs per animal over the multiple days of imaging. We have clarified this in the methods.

8. Methods line 601: Is the procedure described here for pipette insertion/ACSF the same as what is used for loose patch recordings (line 571). If so, these details should be provided in the section on loose patch recordings as well.

Yes, they were the same microelectrodes. To avoid confusion, we now unify the term for both as microelectrodes in the revised manuscript. Moreover, we added the statement "This

procedure was also used for local application of isoflurane in AuC L2/3.” in the loose patch recordings section in the Methods.

9. Methods line 678: Why do the authors use Bayes factor testing for the data in Fig. 4? What is the rationale for choosing this approach here?

The key idea here is to be able to gain information to tell apart ‘absence of evidence’ and ‘evidence of absence’, so as to assess how much off-target effect does our isoflurane application exert to the responsiveness of non-HB cells. The Bayes factor testing offers an approach for this purpose.

Reviewer #2 (Remarks to the Author):

The manuscript from Li et al. entitled "Holistic Bursting Cells Store Long-term Memory in Auditory Cortex" described results from a series of experiments to investigate how auditory cortical neurons establish and store behavioral relevant sound information during learning. Chronic 2-p Ca²⁺ imaging was used to reveal emerged holistic bursting cells (HBCs) in layer 2/3 of mice auditory cortex during an auditory associative learning. Reversal and dissociation of the learning, together with the isoflurane-mediated silencing of these neurons dampened task performance supported the causal effect of HBCs in the auditory associative memory.

The experiments were logically well-designed, and a series of sophisticated techniques were employed. Results are convincing and the finding is significant.

We thank the reviewer for their positive assessment and have addressed all the minor concerns below.

I have some minor concerns detailed below:

1. The recordings were performed focusing on the layer 2/3 of the auditory cortex. It does not rule out the possibility that other neurons in deeper layers are also involved in the learning and memory storage. Isoflurane manipulation may (most likely) also silence neurons in the deeper layers as well. Discussion of this possibility is needed and conclusions need to be precise accordingly.

We agree. We do not rule out the possibility that the isoflurane manipulation may also affect neurons in the deeper layers, specifically those which also possess a high-frequency bursting response property (e.g., de Kock et al, Commul Biol 2021, showing a small fraction of layer 5 thick-tufted cells in barrel cortex exhibiting ~100Hz response to whisker stimuli). We added a discussion right after the conclusion of the relevant data (now Fig. 5 in the revised manuscript).

2. With the similar notation, the claim of "...HB cells as long-term memory storage in the sensory neocortex." is too broad. The result in this manuscript is only supporting the conclusion for the auditory cortex.

We agree and have rewritten both the abstract and main text to align with this point.

3. In Figure 4 the analyses of Ca²⁺ signals showed that isoflurane delivery significantly silenced HBCs but not for other cells. However, since isoflurane is believed to act on the neurons via the same mechanism, it is highly possible that this manipulation is a general suppression of the auditory cortex (including the above threshold and subthreshold activities may or may not detected by the Ca²⁺ imaging analyses). This is important to reach the causality conclusion of the HBCs in this associative memory. This limitation should be discussed.

We agree that the local aqueous isoflurane application will spread to a relatively large volume that includes deeper layers (layers 5/6). We also agree that subthreshold activities could be altered by isoflurane and are not detectable by Ca^{2+} imaging in our experiments. However, as only suprathreshold activities can effectively transmit information to downstream targets (i.e., those brain regions more closely involved in making a behavioral action), we consider only suprathreshold activities in interpreting the behavioral effect of isoflurane application. This limitation is now discussed in the manuscript.

4. It is not mentioned in the text or figures about the GCaMP6m vector. Which promoter was used? Are both excitatory and inhibitory neurons labeled and imaged? Are they separated in the analyses?

We used the hSyn vector of the GCaMP6m, so we could not separate excitatory and inhibitory neurons. We updated the statement of virus in Method as ‘The inclined electrode (diameter: $\sim 20 \mu\text{m}$, Angle: 65° to the horizontal) filled with $\sim 80 \text{ nl}$ AAV2/9-hSyn-GCaMP6m-XC-WPRE virus was slowly inserted underneath the surface of AuC pia’.

5. It is not clear why in figure 3 a different task was used to assess the dissociation effect. This is an auditory detection task and the other used in Figs 1/2/4 is an auditory discrimination task. The auditory cortical neurons may or may not function in the same way on these tasks. The limitation should be discussed.

Indeed, Reviewer #1 also made this point. In principle, any defined composite sound of a constant waveform can be used for this experiment (association \rightarrow dissociation \rightarrow re-association); they don't have to be one of the two chords used for the other experiments (Figs. 1/2/4, now Figs. 1/2/5 in the revised manuscript). Note also the BBN used here was also a specific composite sound by composing a broad band of pure tones with randomized phases and amplitudes and its waveform was ‘frozen’ for all animals and all trials, i.e., not a different noise from trial to trial, but always the same waveform per each trial. Since in the previous study (Wang et al, NC 2020) we have already performed the chronic imaging experiments through a learning cycle till the third phase (dissociation), here in this study we carried on with the same experimental design (using the same ‘frozen’ BBN) and added a re-association phase, for which, 5 mice were imaged in a pilot experiment that was later pooled together with a new round of experiments (10 mice) for Fig.3d (altogether 15 mice). We briefly mention this issue by citing our previous work in the introductory part of this dataset.

Reviewer #3 (Remarks to the Author):

The manuscript by Li et al. is exciting and timely conceptually, technically impressive, and appropriately grounded in a deep and rich literature in the neurobiology of memory. The authors report a physiological signature of auditory associative learning in the auditory cortex, specifically in "bursting" responses of select layer 2/3 neurons (though presumably this kind of bursting activity is possible in other layers, but undetected using the approach in the current report). Strikingly, several tests of the nature of the putative physiological signature of learning show characteristics that qualify these burst responses as an "engram" of the salience of a specific sound chord that is paired with water reward. The bursting cells emerge with learning, survive reversed behavioral contingencies (likely due to the salience of the sounds being unchanged), are long-lasting (up to a month), track extinction by reducing in response activity with reduced sound salience, and (at least in part, some of the cells) reinstate bursting once the chords are again made behaviorally salient.

This is a remarkable set of findings that open the door to many novel questions that the authors may address: For example, what is the circuit function of these layer 2/3 cells? In other words, what cells are downstream of this activity? Upstream? Is there any predictable topography of these cells based on, say, the acoustic frequencies of the salient chords? At a molecular level, novel questions emerge for the single-cell: What are the cellular consequences of burst activity with respect to gene expression, calcium-related signalling cascades, etc., that could predict persistent changes to cellular functions that support bursting activities?

The authors go on to perform a causal experiment using a clever reversible manipulation using isoflurane to decrease bursting activity without obliterating responsiveness more generally across the auditory cortex. Remarkably, the selective decrease in bursting blunts the behavioral performance of the task, which returns to high performance levels after wash-out. While it is difficult to interpret the specificity of the change in behavior to only this task and chord stimuli, the result is nonetheless compelling as a test of "necessity" of these cells' bursting activity to maintain the behavior, particularly in the context of the other behavioral manipulations reported in the manuscript. Though beyond the scope of the present report, it would be interesting to hear the authors speculate on whether or not bursting activity is "sufficient" to drive cued behavior. Necessity and sufficiency of these layer 2/3 "bursting" responses would definitively identify them as engram cells in the Semon (1921) definition.

Thank you very much for your appreciation of our data and for your insightful suggestions.

To your question, whether or not bursting activity is "sufficient" to drive cued behavior, we unfortunately have not made a specific causal test yet (e.g., holographic targeted optogenetics activation of identified HB cells), however, the situation is more complicated than a simple causal relationship. Here (additional data, figure panels below), during one experimental day, the trained animals could lick either upon sound stimulation at a high chance or spontaneously at a low rate (panel a), however, HB cells did not fire bursts in those spontaneous licking events in the absence of sound stimulus (panels b, d). Moreover, the burst firing response of HB cells are so reliable (almost 100%) with a latency much ahead of the licking action onset. Therefore, a 'gain-of-function' (sufficiency) test in the presence of a learned composite sound as a cue is not really relevant, because the HB cells will fire bursts anyways at near 100% probability upon their specific learned sound (that's also why we focused on the 'loss-of-function' test in this study). Crossing these two correlational observations will naturally lead to a conclusion that the sufficiency can't be easily rejected, particularly in view of the tight timing relation between sound stimulus, HB cell burst firing and behavioral action one after the other. The only alternative interpretation is that somewhere in the upstream auditory processing pathway the memory information is already retrieved and activating the behaviors, while HB cells get a copy relay. For this alternative interpretation we also had a pilot dataset that can partially reject it, i.e., a caspase-mediated L2/3 neuronal ablation experiment (unpublished observations in pilot).

Nevertheless, sufficiency can still be studied in the future, but rather in the aspects of holism and dormancy. First, we could present the animal with a pure tone stimulus (like those in the holistic tests, Fig.1) and in the meanwhile specifically drive a selected set of HB cells (for one of the two learned chords) to fire bursts (which would otherwise not fire bursts for the tone, per definition of HB cells), to see whether or not in this case the animal would significantly raise the correct behavioral response level (a tone could elicit 20~40% licking, in contrast to a learned chord of almost 100%, note that this also means the first choice on left/right is almost 100% correct). Furthermore, in the re-association experiments (Fig. 3), we could also drive those HB cells which become quiescent through the dissociation training and see whether or not the behavioral response would fully recover already in the first day of re-association. However, we must first pass some technological feats to perform these experiments – including developing a new microscope, which is needed in the context that our new data (new Fig. 4) shows HB cells are very sparsely and widely distributed throughout both primary and secondary auditory cortical areas and our current holographic all-optics stimulation device cannot cover such a large zone. A brief version of this explanation has been added to the last paragraph of the discussion in the manuscript.

On the matter of sufficiency: To what extent to layer 2/3 cells burst

spontaneously? Do the authors have inter-trial data that could address whether bursting is only sound-evoked?

We have performed a new set of experiments for the revised manuscript to specifically address this question. The new data are appended to Fig. 1 and Suppl. Fig. 1, where we performed recordings of spontaneous activity in long time windows without any sound stimulus. In both groups of non-HB cells and HB cells, the spontaneous bursting event rate was comparably low. In contrast, the probability of a bursting response evoked by the specific learned chord in HB cells is almost 100%. Moreover, the number of spikelets of a learned chord-evoked burst event is significantly more than that of a spontaneous burst event in either non-HB cell or HB cell (please see detailed numbers in the main text and updated figures).

On a separate note, I did not see the performance calculation(s) used to quantify behavior across the different tasks and task phases in the methods or figure legends. These are important to include to appropriately interpret learning (acquisition), memory, and task performance abilities.

We apologize for not explicitly citing this information in the methods section. We extracted the lines of method description for behavior data analysis and made a separate paragraph, as follows:

Behavioral data analysis (all figures)

We used custom-written code in MATLAB to extract the tongue movements and determine the timepoint when the tongue touches a water spout (left or right). A trial was defined as a success if the animal's first lick touched the correctly assigned spout within 1000 ms following the sound presentation. Importantly, in case a spontaneous licking event coincidentally started before the end of the sound stimulus, the trial was discarded (but no punishment was given to the animal) and did not contribute to the denominator for the calculation of success rate. The behavioral analysis included all valid trials from the beginning of the session until the moment when altogether 20 droplets of water had been consumed. If the animal did not consume the water reward before the next trial began, a new droplet would replace the existing droplet. Usually in the 2-chord scenario, animals tended to lick on both spouts one after the other, but we only considered the first hit being on the correctly assigned spout as a success trial, and there was no punishment applied to the animal if it licked first on the wrong side.

Reviewer #4 (Remarks to the Author):

In this manuscript, the authors report that neurons in the auditory cortex exhibit plasticity changes after sound cue associative learning. Although I am quite interested in the topic of the study, the results and claims of the study are quite confusing. The study has substantial overlaps of results with the previous publication from the same group titled "Single-neuron representation of learned complex sounds in the auditory cortex (Wang et al, nature commun. 2020)". In addition, the claimed mechanism for "long-term memory storage", or the claimed holistic responses are not supported by the experimental evidence.

Thank you very much for your interest in our studies. In fact, except for half of the figure 1 which verifies the observation of HB cells in the previous paper (Wang et al, NC 2020) by a directly targeted loose-patch recording following chronic imaging (instead of a calibrated acute Ca²⁺ imaging), all the rest of this manuscript (now 5 figures in the revised version) contain new experiments and new findings that provide multiple lines of converging evidence that the HB cells embody a memory engram of the learned, behaviorally relevant composite sounds. With all due respect, the overlap is only minimally required for the consistency of science, and not at all 'substantial'. Therefore, our new data both replicate fundamental previous findings (not always trivial) and also provide novel information and an advancement in our knowledge. We also argue that our findings can indeed support our main conclusion, but we have adapted our comments to cautiously interpret and discuss our data for drawing this conclusion in the revised manuscript. We hereby address your concerns point by point.

1. Previous studies have shown that classical and operant conditioning can lead to a global shift in the tonotopic map toward the frequency of the conditioned sound in the auditory cortex, and that auditory training can alter the tuning properties of individual neurons in the auditory cortex. It's well established that learning can lead to changes in the neural representation of sound in the auditory cortex. The authors should consider these findings when interpreting their results.

Thank you for raising this issue. We performed a new set of experiments (new Fig. 4) and added new discussions to further explore the relationship of our data to the tonotopic representation. We performed a new combined experiment with wide-field epifluorescence Ca²⁺ imaging (large-view mapping, no single-cell resolution) and 2-photon Ca²⁺ imaging (small view, single-cell resolution and high sensitivity) in the same post-training animals sequentially (both using the same GCaMP6m labelling). Note that, instead of pure tones used in the majority of literature, we trained the animals with either specifically composed multi-tone chords or BBN with a constant waveform, hence, there is not necessarily a direct comparison to be made in relation to changes in frequency tuning per se (therefore, we have not done so). It may be specifically relevant that HB cells are selective to composite sounds and not necessarily their constituent pure tones. After all, real-world behaviorally relevant sounds each have a unique characteristic

composition of multiple tonal components, in contrast to the fact that single pure tones are rarely presented in naturalistic life. Nevertheless, we addressed the relation between the tonotopic map and the emerged HB cells. Please see the new Fig. 4 and the data therein.

2. The recorded auditory cortical responses seem weird. Responses to pure tones are not observed or extremely weak even at around 80dB. If the majority of the neurons did not exhibit robust responses, their comparison with and without isoflurane would not be meaningful. The authors should consider the possibility that the tone used may be outside the receptive field of the recorded neurons, rather than the neurons "emerging from quiescence."

There indeed exists cells that exhibit bursting responses to both a learned chord and a constituent tone (referred to as 'quasi holistic bursting' cells, as Fig. 6 in the previous paper, Wang et al, NC 2020) but those are the minority as compared to the complete wholistic ones (bursting to a chord only, not to any constituent tone). Note that our question in focus here is whether the bursting outputs of HB cells are required for a memory engram to enable behavioral action, and the isoflurane experiment was designed for that. Please also refer to the previous question, that the entire study paradigm is not much relevant to the classical tone-tuning response and tonotopic maps of the auditory cortex. Specifically, the 8 tones that were used to compose the two chords span a broad range of the audible spectrum of mice, and thus there is not much chance that receptive fields of HB cells would not be overlapping with any of those. As above, please see also the new Fig. 4 and the data therein in relation to the location of HB cells in the auditory cortex.

3. The specificity of the persistent plasticity in the selected population has not been convincingly demonstrated, as other recorded cell types have not been included in the comparison. It is essential to address whether other cells in the auditory cortex might also exhibit plasticity after sound-related learning, which seems not consistent with previous experiment results.

Our study does not preclude that plasticity is also ongoing in neurons that respond to pure tones as has been previously described in literature, however, the nature of the HB cells themselves, responding in a bursting manner specifically to composite sounds, may suggest that they serve a unique role in processing and remembering more complex, naturalistic, and behaviorally relevant sound stimuli. We have appended a discussion in the revised manuscript to address your concern.

4. The loss-of-function experiment presented in the manuscript is not convincing. What's the foundation of the specificity that systematic and local application of isoflurane specifically silenced the observed cells while sparing all other cell types, layers, or regions.

The specific suppression of high-frequency activity is founded mechanistically in the cited paper (Wang et. al, J Neurosci 2020, Frequency-Dependent Block of Excitatory

Neurotransmission by Isoflurane via Dual Presynaptic Mechanisms). The findings of that paper suggest that the low-pass filtering effect of isoflurane was very suitable for specifically suppressing the high-frequency (at the order of 100 Hz) burst firing of HB cells while allowing low-frequency firing non-HB cells to continue to fire. We are confident that the same biophysical mechanism applies in our study.

Note that our conclusion is based primarily on the local application experiment and the systemic application served only for the dosage and cellular mechanistic control test (see also Suppl. Fig. 3). Our new Fig.4 shows that HB cells are widely distributed in not only primary but also secondary auditory cortical areas, and thus the lateral regional diffusion and spread of isoflurane beyond the injection volume (visualized with Alexa-594) is favorable. A minor concern is that some cells in the deeper layers, specifically those which also possess a high-frequency bursting response property (e.g., de Kock et al, *Commul Biol* 2021, showing a small fraction of layer 5 thick-tufted cells in barrel cortex exhibiting ~100Hz burst firing response to whisker stimuli) may also be affected by the spread of isoflurane, though the bursting capability of those deep layer cells are not necessarily emerged through learning but are known to be naïvely existing. We added a discussion in this regard right after the conclusion of the relevant data (now Fig. 5 in the revised manuscript).

5. The non-specific silencing of the auditory cortex resulting in a deficit of auditory-associated learned behavior should not be surprising. This may not be considered as a supporting evidence for long-term memory storage.

Both literature and our data show that there is a certain degree of specificity of the local isoflurane application that preferentially suppresses the high-frequency output of the HB cells in layer 2/3 (with a possible off-target effect in deeper layers where high-frequency bursting responsive cells also exist, albeit those cells' bursting responsiveness are not necessarily learning-transformed). We agree that the interpretation of the isoflurane experiment must be cautious (see newly appended discussion texts in the revised manuscript) as the other reviewers are also concerned, but discrediting the data as not being supporting evidence for memory storage is unfair and deviates from the consensus of the other reviewers.

6. What's the basis for the so called "holistic" response? The observation could be explained by the integration function of the recorded neurons being non-linear (e.g., exponential) rather than truly "holistic."

We fully agree with your explanation, because what you refer to as 'non-linear integration' (supralinear, if we understand correctly) means for us 'the whole (learned chord response) is greater than the sum of parts (constituent tone response)', i.e., the literal meaning of 'holistic' (=wholistic). However, without a future study to dissect the input-output relation within the specific single cells we cannot assess the specific form of non-linear integration function here, thus a literal term 'holistic' here is justified.

7. The observed responses could be the responses induced by conditioned reward rather than the holistic response to complex sound. The responses

correlated better with reward/behavior performance, rather than the complex sound per se, and the response decreases with decrease in behavior performance, and vice versa. Once trained without water reward, the response was largely dismissed.

Thank you very much for the interesting question. We may kindly reject the interpretation that these are only ‘reward’ responses with a simple timing relation (see here the attached data panel, also Fig. 3e, 3g in the previous paper Wang et al, NC 2020) showing that the onset of sound-evoked bursting response precedes either the reward being dispensed (blue shade in the figure) or the licking action onset.

However, apart from the timing issue for judging the interpretation, we fully agree with the provided abstracted description of our data, as the bursting responses do not merely convey the information of the composite sound (indeed they only emerge with learning), but also convey a behavioral relevance to downstream recipients which are likely more directly accountable for conducting the actual behavioral action. Here in lies the fascinating and unique property of these HB cells and their relationship to maintaining a long-term memory trace of behaviorally-relevant composite sounds in auditory cortex!

REVIEWERS' COMMENTS

Reviewer #1 (Remarks to the Author):

In their revision of the manuscript “Holistic Bursting Cells Store Long-term Memory in Auditory Cortex,” the authors significantly strengthened the paper via additional experiments, analyses, and discussion of relevant caveats. In particular, the authors performed additional loose patch recordings to strengthen the validity of the “calibration curve” for tagging HB cells, and performed additional experiments and analyses examining the distribution of HB cells throughout auditory cortical areas (new Fig. 4). The authors added important discussions of the caveats of tagging the “putative” HB cells based on the calibration curve, and softened the interpretation of the changes (or lack thereof) in HB cells in response to reversal training. Finally, the authors clarified the few points of confusion in the methods.

Overall, my enthusiasm for this manuscript and its significance remains strong. It establishes a model of memory “engram” cells in sensory cortex in a real-time imaging setup during learning, sheds new light on possible mechanisms of memory, and develops techniques for studying these mechanisms. My comments and concerns have been addressed.

Reviewer #2 (Remarks to the Author):

The revision and response have addressed all my concerns.

Reviewer #3 (Remarks to the Author):

The authors have done an absolutely excellent job in responding to the broad reviewer comments, suggestions and requests. The existence of sparse HB cells across the auditory cortex warrants future investigation to dive deeper into their functional role in the cortical circuit. This paper (combined with the 2020 paper from the same group) is a fantastic starting point to initially characterize this functional cell type.

Reviewer #4 (Remarks to the Author):

The manuscript has shown some improvements; however, my major concerns still persist. The study remains in a preliminary stage as evidenced in several areas:

1. The primary claim of the study—that H-cells store long-term memory—relies heavily on evidence indicating selective effects of isoflurane (a general anesthesia reagent) on neuron's bursting responses. However, this evidence remains tenuous, and the interpretation of both the literature and the presented data appears somewhat arbitrary. There is also no clear mechanism support the claimed specificity.
2. Regarding the potential overlap with the authors' previous publication in Nature Communications: Although new data are introduced, the major point on the H-cell responses were already fully explored in the previous paper. While the current work appears to be a continuation of that study, this connection has been seemingly sidestepped, with no acknowledgment or discussion.
3. From a logical standpoint, the study's experimental design does not bolster its main claim. While suppression experiments might suggest these neurons' crucial role in memory-related tasks, they don't elucidate the exact nature of this contribution. To say they store memory feels like an overreach based on the data presented.

More specifics on isoflurane experiments:

The most pivotal evidence supporting the primary claim is the selective silencing of holistic neurons using isoflurane. Yet, neither the extant literature nor the authors' own data supports the claimed specificity in this study. For instance, Wang's 2020 study (Figures 8F-G) indicates a significant suppression in both spontaneous and evoked neuronal activity at low frequencies of 0.2 and 2Hz. This interpretation seems at odds with the present study. Furthermore, Hentschke's 2017 research didn't specifically focus on individual neurons, but rather on multiunit, population-level activities. They reported an overarching response suppression, where both early and late sound-evoked responses were reduced by 50% to 100%.

Moreover, the authors' data don't align with existing literature or the current understanding of auditory cortical responses. There's notable concern regarding the overall exceptionally low evoked responses from NH neurons in this study. A substantial portion of ACx neurons should exhibit sound-evoked responses, and only sound responding neurons should be applied in this analysis. If these low responses are due to calcium imaging signals, then it isn't a reliable measure of auditory cortical responses.

Additional single-cell ephys experiments would be essential to validate these findings. To more definitively characterize effects on sound-evoked responses, using varied frequency/intensity sound stimuli would be advisable.

We thank all the reviewers for their highly constructive comments both in the past and herein for helping to improve our manuscript. Here, we address their new concerns point-by-point.

Reviewer #1 (Remarks to the Author):

In their revision of the manuscript “Holistic Bursting Cells Store Long-term Memory in Auditory Cortex,” the authors significantly strengthened the paper via additional experiments, analyses, and discussion of relevant caveats. In particular, the authors performed additional loose patch recordings to strengthen the validity of the “calibration curve” for tagging HB cells, and performed additional experiments and analyses examining the distribution of HB cells throughout auditory cortical areas (new Fig. 4). The authors added important discussions of the caveats of tagging the “putative” HB cells based on the calibration curve, and softened the interpretation of the changes (or lack thereof) in HB cells in response to reversal training. Finally, the authors clarified the few points of confusion in the methods.

Overall, my enthusiasm for this manuscript and its significance remains strong. It establishes a model of memory “engram” cells in sensory cortex in a real-time imaging setup during learning, sheds new light on possible mechanisms of memory, and develops techniques for studying these mechanisms. My comments and concerns have been addressed.

Reviewer #2 (Remarks to the Author):

The revision and response have addressed all my concerns.

Reviewer #3 (Remarks to the Author):

The authors have done an absolutely excellent job in responding to the broad reviewer comments, suggestions and requests. The existence of sparse HB cells across the auditory cortex warrants future investigation to dive deeper into their functional role in the cortical circuit. This paper (combined with the 2020 paper from the same group) is a fantastic starting point to initially characterize this functional cell type.

Reviewer #4 (Remarks to the Author):

The manuscript has shown some improvements; however, my major concerns still persist. The study remains in a preliminary stage as evidenced in several areas:

1. The primary claim of the study—that H-cells store long-term memory—relies heavily on evidence indicating selective effects of isoflurane (a general anesthesia reagent) on neuron’s bursting responses. However, this evidence remains tenuous, and the interpretation of both the literature and the presented data appears somewhat arbitrary. There is also no clear mechanism support the claimed specificity.

We agree that the loss-of-function experiment by using local nano-injection of aqueous isoflurane in awake behaving animals (not as general anesthesia in the way as the reviewer interprets) raises concerns and disputes in its interpretation. Therefore, for publishing this piece of solid experimental data with the already-included discussions on the concerns of this experiment, **we tune down our conclusion as H-cells are specific single cell candidates of auditory engram underlying long-term**

memory storage in the auditory cortex, and we remove the wording ‘causal’ in the interpretation of isoflurane experiment. In particular, we would like to kindly remind that there are 3 other independent lines of evidence on the specificity, namely the content (Fig. 1), the persistence (Fig. 2) and the dormancy (Fig. 3) that are all strongly supportive of the notion of a memory engram as originally defined by Richard Wolfgang Semon since 1905. As such, our primary claim of the study does not heavily rely on that one disputable experiment.

2. Regarding the potential overlap with the authors’ previous publication in Nature Communications: Although new data are introduced, the major point on the H-cell responses were already fully explored in the previous paper. While the current work appears to be a continuation of that study, this connection has been seemingly sidestepped, with no acknowledgment or discussion.

We thank the reviewer for their suggestion of explicitly elaborating the connection between the current study and our previously reported study. We have revised both the introduction and discussion sections accordingly, and as demonstrated, **the present study is not merely a continuation of the previous study but brings forward a major conceptual advancement, i.e., demonstrating ‘holistic bursting’ cells in the auditory cortex not only represent the learned composite sounds (previous study), but also invariably persist with this specific representation and could reinstate from dormancy (this study)**. Even without the disputable isoflurane experiment, these findings together directly pin down one specific physiologically-defined class of cells in the neocortex as substrate of memory storage. In contrast, concurrent engram tagging methods using immediate early gene (IEG) expressions are showing indirect proxies of neuronal activity underlying memory formation – which by the way, inexplicably does not work for the sensory neocortex (where sensory-related neuronal activity timescales are much faster than gene expressions). It is true that we conceptually reproduced our own finding of the H-cells (previous study), but our reproduction (this study) is with new experiments of much better precision and richness, including a systematic comparison between sound-evoked and spontaneous activities in loose-patch recordings (Fig. 1) and a brain-wide map of the H-cells (Fig. 4). **We appended the relevant information in the introduction section of the current version of manuscript for the readers to better understand the connections between the past and the present.**

3. From a logical standpoint, the study's experimental design does not bolster its main claim. While suppression experiments might suggest these neurons' crucial role in memory-related tasks, they don't elucidate the exact nature of this contribution. To say they store memory feels like an overreach based on the data presented.

We acknowledge the presence of the reviewer’s interpretation of this piece of isoflurane intervention data, and we do not intend to argue here despite we can address these concerns technically. All the other 3 reviewers have not made further comments regarding this issue, as we have already presented the relevant concerns in a fair manner in the previously revised manuscript. Thus, we leave this experiment to an open discussion for the readers, and we also leave for ourselves an open avenue of technical development on cortex-wide single-cell silencing to address this question in the future (also mentioned in the current re-revised manuscript). After all, **our conclusion for the current study is tuned down such that ‘holistic bursting’ cells are the specific single-cell candidates of long-term memory storage in auditory cortex (accordingly, the ‘causal’ wordings throughout the manuscript are removed).**

More specifics on isoflurane experiments:

The most pivotal evidence supporting the primary claim is the selective silencing of holistic neurons using isoflurane. Yet, neither the extant literature nor the authors' own data supports the claimed specificity in this study. For instance, Wang's 2020 study (Figures 8F-G) indicates a significant suppression in both spontaneous and evoked neuronal activity at low frequencies of 0.2 and 2Hz. This interpretation seems at odds with the present study. Furthermore, Hentschke's 2017 research didn't specifically focus on individual neurons, but rather on multiunit, population-level activities. They reported an overarching response suppression, where both early and late sound-evoked responses were reduced by 50% to 100%.

Despite we do not agree on these specific technical issues on interpreting data in literature and comparisons with our study, here we will not argue because we already agreed to set aside the minor dispute in interpreting the isoflurane experiment.

Moreover, the authors' data don't align with existing literature or the current understanding of auditory cortical responses. There's notable concern regarding the overall exceptionally low evoked responses from NH neurons in this study. A substantial portion of ACx neurons should exhibit sound-evoked responses, and only sound responding neurons should be applied in this analysis. If these low responses are due to calcium imaging signals, then it isn't a reliable measure of auditory cortical responses. Additional single-cell ephys experiments would be essential to validate these findings. To more definitively characterize effects on sound-evoked responses, using varied frequency/intensity sound stimuli would be advisable.

The reviewers' understanding of auditory cortical responses are very accurate, convincingly showing that the reviewer is an expert in the auditory physiology field. However, it is rather the misunderstanding of our experiments with new paradigm and methodology that led to the conflict. Here, please allow us to elaborate:

(1) Our specific sound stimuli (composite chords) have a very short duration of 50 ms. Non-bursting neurons could have a chance to fire maximally a few singlet spikes in and around such a short time window, however, it is well known that most ACx L2/3 neurons respond unreliably to naive sound stimuli of whichever kind (Bandyopadhyay et al, Nat. Neurosci. 2010).

(2) A striking finding in our study is that single L2/3 ACx neurons transformed from quiescence to strongly bursting responsive through learning. Our experiments primarily involve chronic imaging which could detect both silent neurons and active neurons in any stage of the learning, we do not understand why we should restrict the analysis to 'only sound responding neurons' (e.g., only those detectable with blindly inserted microelectrodes).

(3) These low responses of NH neurons are unlikely due to calcium imaging signals, because in both this study and our previous study we performed simultaneous loose-patch and two-photon calcium imaging experiments to 'calibrate' the calcium signals and show that our experiments are sufficiently sensitive to detect low-rate firings.

(4) A great deal of imaging-guided loose-patch experiments have been included in the previously revised manuscript and all other 3 reviewers are satisfied with them. We do not understand why more of such data is needed, since in each animal we can only penetrate the cortex a limited number of

times with patch pipettes, yet any unnecessary use of animals is against ethical rules. The combination of calcium imaging and single-cell loose-patch is a highly suitable method to address this paradox of low volume of high-precision data.

(5) To the very specific point, our design of learning with such composite multi-tone chord stimuli was advised by highly renowned expert of auditory physiology. Our aim is not to characterize sound-evoked neuronal response patterns since there have been already numerous studies of such purposes in the past, regardless of using pure tones or tone sweeps or other sound patterns as sound stimuli. Our aim is to pinpoint which exact single cells (out of a big population) are candidates of engrams underlying long-term auditory memory storage in ACx – given the other line of literature suggesting that ACx as a large brain region possesses traces of auditory memory (e.g., Weinberger et al, Nat Rev Neurosci 2004).